# Person Detection Through the Lens of Algorithmic Bias

## Abstract

The rise of AI based person detection in safety critical applications such as driverless cars or security monitoring has lead to an explosion of machine learning models and dataset research. At the same time, researchers have raised question of bias in these models and datasets. Popular benchmark datasets like More Inclusive Images for People (MIAP) and Berkeley DeepDrive (BDD) for person detection suffer from both sampling and labeling biases. This has serious implications for autonomous vehicles and other fields that use these datasets. We conduct an all-encompassing analysis to assess these datasets through the lens of algorithmic bias, looking at both dataset and model bias. To the best of our knowledge, no study has delved into the realm of person detection in low-quality or crowded pictures with this lens. The result is a novel analysis of bias in a real-world image dataset. We find that 1) image manipulations frequently found in real-world settings like image blurriness and 2) image detectors that are skewed to rely on features like contrast or brightness both have significant negative impacts on fairness for race, gender, and age demographics. These result can help guide future designs of robust models in the object detection field and beyond.

## 1 Introduction

Machine learning (ML) has brought about a revolution in computer vision (CV). Tasks that were once considered labor-intensive or even infeasible have become achievable through the progress of CV research. Object detection, an important branch of CV, involves both locating an object within an input image and classifying it. This is a fundamental task for ML-powered video surveillance, autonomous vehicles, facial recognition systems and pedestrian detection (Fabbrizzi et al., 2022).

While modern object detection models exhibit a high degree of precision, they are not impervious to biases within the data and models themselves. These inherent biases can influence the performance and fairness of object detection outcomes (Fabbrizzi et al., 2022; Wang et al., 2019b;a; Hazirbas et al., 2021; Goyal et al., 2022; Agarwal et al., 2021; Wang et al., 2020b; Meister et al., 2023). Specifically, research has revealed disparities in the accuracy of person detection models for individuals with darker skin tones or those partially concealed by other objects (Wilson et al., 2019). In the context of autonomous vehicles, which have been involved in several high-profile pedestrian deaths since 2018, minimizing bias is absolutely critical for public safety (Levin & Wong; Biddle, 2023; Kerr, 2023; Cummings, 2023; Valdes-Dapena, 2024). Despite this, there is worrying evidence that pedestrian detection systems do contain noticeable bias (Bae & Xu, 2023; Althoupety et al., 2023; Katare et al., 2022). Therefore, it is imperative to conduct systematic experiments into how object detection models behave in the context of human detection, while simultaneously considering biases present in both the data and the models.

## 2 Problem Statement

Object detection is a ubiquitous task found across various domains, including agriculture, safety, and retail. However, like any machine learning technique, object detection methods can inherit biases that might disproportionately affect marginalized populations. Additionally, annotations can reflect the biases of the people who produce them. Consequently, it is imperative to comprehensively assess

these methods and the data fed into them for potential bias. While most of the works reviewed in Section 2 focused on bias detection and mitigation, prior research has mainly revolved around gender detection or portrait-like imagery. To the best of our knowledge, no study has delved into the realm of person detection in low-quality or crowded pictures. In this study, we employ two datasets and two object detectors to explore how biases in the image dataset and object detection models can impact the outcomes of the object detection system with respect to fairness and performance.

## 3 BACKGROUND

### 3.1 OBJECT DETECTION TECHNIQUES

Real-world applications of object detection include autonomous driving and video surveillanceZou et al. (2019). Traditional detectors, such as the [HOG Detector] (Dalal & Triggs, 2005), were built on manually constructed features and lacked access to accurate image representation. The resurgence of CNN's popularity prompted researchers to implement this technology in the field of object detection (Zou et al., 2019). In this work, Faster RCNN (Ren et al., 2015) and YOLO (Redmon et al., 2016) will be used to examine the performance of both types of CNN-based object detection methods when there is a bias in the data or model, since these two architectures are prevalent in the literature.

### 3.2 FAIRNESS METRICS

In this study, fairness definitions will be used to evaluate the performance of several object detection techniques. Below, we discuss the most common fairness definitions in machine learning research: Kusner et al. (2017); Dwork et al. (2012); Mehrabi et al. (2021); Corbett-Davies & Goel (2018); Corbett-Davies et al. (2017); Hardt et al. (2016); Verma & Rubin (2018). In the ML literature, fairness can be categorized as statistical fairness measures, similarity-based fairness measures according to Verma and Rubin Verma & Rubin (2018).

Statistical conceptions of fairness serve as the foundation for assessing the fairness of machine learning models utilizing the ground truth label and predicted outcome Verma & Rubin (2018). Typically, these measurements are generated directly from the model outcome or confusion matrix utilizing statistical metrics such as True Positive (TP), False Positive (FP), True Negative (TN), False Negative (FN), Positive Predictive Value (PPV), True Positive Rate (TPR), False Discovery Rate (FDR), and False Negative Rate (FNR) (see Appendix A.1 for definitions). Statistical fairness measures are predicted based on outcomes and ground truth labels which are separated into positive and negative categories. Depending on the context, positive and negative are defined differently. For example, in the case of CT scans, the positive class could refer to malignant cases, whereas the negative class refers to benign cases. In this dissertation, only statistical measures that apply to the object detection space and do not typically include negative examples will be used, as negative examples (images without specific classes) are not fed into the object detection pipeline.

Statistical methods like Statistical Parity (SP), Equalized Odds (EO), and Equal Opportunity (EOP) assist researchers in quantitatively quantifying the results of ML models (see Appendix A.2. for definitions). Their key advantage is that they are simple to compute. Nonetheless, these statistical definitions may contradict one another, and there is no uniform definition applicable in every circumstance. In addition, they may not be suitable in applications that lack tabular data for identifying positive and negative classes to compute statistical metrics.

Statistical fairness measures analyze only a subset of statistical metrics based on the sensitive data attribute. Another category of definitions of fairness is based on the similarity of the occurrences, assessing whether or not they had similar outcomes. In Appendix A.3, we define several popular similarity-based fairness measures.

### 3.3 RELATED WORKS

Benchmarking datasets like ImageNet Deng et al. (2009), COCOLin et al. (2014), VPASCAL Everingham et al. (2010) were designed to represent the real world and ultimately improve the performance of algorithms. However, this did not happen in the real world, as many of these datasets suffer from the biases, such as labeling, framing, and representation or selection bias. For instance, in MS

COCO Lin et al. (2014), women occur in more indoor scenes and kitchen objects, whereas men co-occur more with sports objects and in outdoor locations Wang et al. (2020a; 2021). ImageNet Deng et al. (2009); Yang et al. (2020) also suffers from different types of biases. For instance, most of the cars captured in this dataset are racing cars, which are not commonly found on the street.

Wilson et al. Wilson et al. (2019) investigated the inequality problem in object detection by using Berkeley Deep Drive (BDD) Yu et al. (2018) dataset, which contains people with varying skin tones and occlusions. Upon using the Fitzpatrick Skin tone Fitzpatrick (1988) in pedestrian detection space, they noticed poor performance for individuals with darker skin tones (4-6). They designed experiments to find out the possible source of this inequality. Wilson et al. Wilson et al. (2019) aimed to investigate if times of day (night or day), occlusion, or the loss function leads to higher accuracy for the majority group or not. They found that neither time of the day nor occlusion are the primary source of the discrepancies. They were able to improve the disparities for dark skin tone individuals by reweighting the loss functions carefully.

Balakrishnan et al. Balakrishnan et al. (2021) developed an experimental method to measure the algorithmic bias of face analysis algorithms by directly manipulating the attributes of interest to reveal the causal links between attribute variation and performance change. They aimed to measure fairness to reveal causal connections between attributes of interest and algorithmic performance. They created synthetic test images using StyleGAN Karras et al. (2019) and identified their gender using annotators and classifiers to detect the biases. They found biases related to gender, hair length, age, and facial hair.

Historically, the performance of a model was measured using accuracy and other statistical metrics. However, there is a need to develop fairness metrics that compute the consistency of model behavior across different groups and attributes. Joo and Kärkkäinen Joo & Kärkkäinen (2020) used an encoder-decoder network for image attribute manipulation to synthesize facial images varying in the dimensions of gender and race. They used these images to compute the counterfactual fairness of commercial computer vision classifiers by examining the degree to which these classifiers are impacted by gender and racial cues.

To the best of our knowledge, our work differs from prior work in fairness in computer vision. Specifically, we investigate algorithmic bias from the lends of the biased datasets and bias detectors on for person detection in low-quality or crowded pictures. The following sections present our form research questions, methodology, results, and discussion.

## 4 RESEARCH QUESTIONS

- **RQ1 (biased dataset):** In object detectors used for human detection, do state-of-the-art visual bias modifications, such as gray scale conversion and blurriness, impact performance and fairness between different demographic groups?

- **RQ2 (biased models):** In object detectors used for human detection, does a biased model based on visual features such as size, contrast, and brightness impact performance and fairness of outcomes between different demographic groups?

## 5 METHODOLOGY

In this section, we describe the datasets used in our experiments. Both were chosen because they contain real-world images and prior work used them for bias research (Wilson et al., 2019).

### 5.1 MORE INCLUSIVE IMAGES FOR PEOPLE (MIAP)

Schumann et al. (2021) released a new subset of the popular Open Images dataset called More Inclusive Images for People (MIAP) has over 100k more annotations than the original Open Images dataset. MIAP is labelled with fair analysis in responsible AI projects in mind. Each annotated bounding box includes age and gender information. In addition to the age and gender labels included with MIAP, skin tone annotation was performed for this study. Each bounding box was extracted from MIAP, passed to Enhanced Deep Residual Networks for Single Image Super-Resolution

(EDSR) to improve the resolution, and had the face detected using the Haar Cascades classifier (Viola & Jones, 2001). The Individual Typology Angle was calculated for each face and the resulting values were mapped to light skin tone (LS), defined as individuals with a Fitzpatrick Scale score of 1 to 3, and dark skin tone (DS) labels, defined as individuals with a Fitzpatrick Scale score of 4 to 6 (Fitzpatrick, 1988). If no face was detected, the entry was labelled "unknown".

## 5.2 BERKELEY DEEPDRIVE (BDD)

Berkeley DeepDrive (BDD) is an annotated dataset intended for object detection in autonomous driving. Wilson et al. (2019) released a smaller version of this dataset consisting of images with skin tone annotations using the Fitzpatrick scale.

## 5.3 ADDITIONAL VISUAL FEATURES

In addition to the human attribute features included in MIAP and BDD, size, contrast, and intensity were computed using the following definitions inspired by the work of Sobti et al. (2021).

- **Size:** The bounding box is small if it is less than 4% of the total image area.
- **Contrast:** Contrast is computed using Equation 1 where $I_{bb}$ and $I_{background}$ stand for the intensity of bounding box and intensity of background, respectively. A bounding box has a low contrast if the contrast value is less than $0.4$.

$$contrast = \frac{I_{bb} - I_{background}}{I_{background}} \tag{1}$$

- **Intensity:** Intensity is computed using average mean values of pixels over all channels. A bounding box is considered dark if the brightness is less than $90$.

## 5.4 BIASED DATASETS

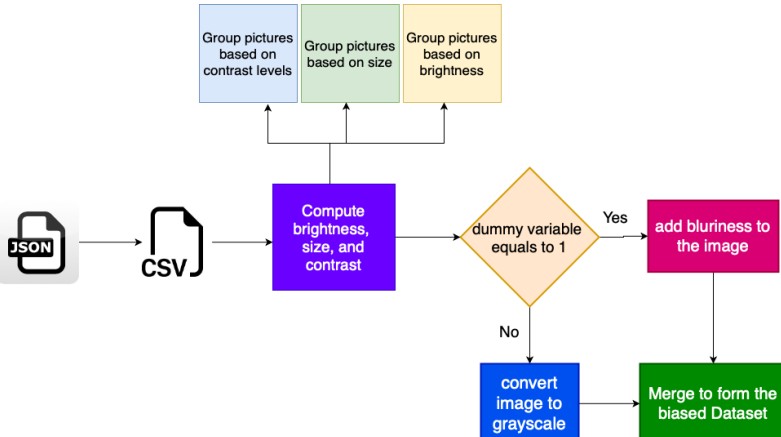

Figure 1: Illustration to show steps to make biased data set concerning size, brightness, and contrast.

While both the MIAP (Krasin et al., 2017) and BDD (Wilson et al., 2019; Yu et al., 2018) datasets inherently harbor certain representation and labeling bias, this study introduces deliberate manual augmentations to inject framing bias and scrutinize its impact on person detection outcomes. Among the various perturbations, the most common practice is to convert images to grayscale or apply blurriness (Wang et al., 2020b). Thus, we consider two bias manipulation techniques: 1. converting images to grayscale and 2. introducing blurriness to bounding boxes containing people. Which augmentation was applied to an image was chosen at random, as seen in Figure 1. Following data preparation, visual features such as contrast, intensity, and size are computed for each bounding box. Grayscale conversion and adding blurriness examples are shown in Figure 2.

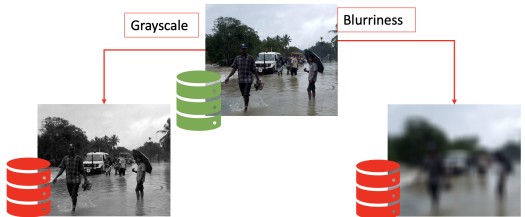

Figure 2: Examples of adding blurriness and grayscale conversion no a picture from MIAP dataset.

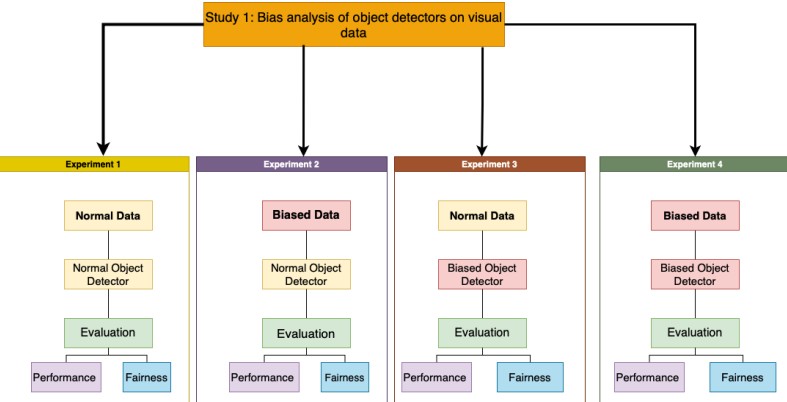

Figure 3: Detailed illustration of the experiment. We use this protocol for datasets and detectors.

## 5.5 SYNTHETIC BIASED DETECTORS

In addition to exploring the biased datasets, this study delves into the effects of biased object detectors. Two base object detectors were used: You Only Look Once (YOLO) and FasterRCNN (FRCNN). The YOLO implementation is based on the Keras-YOLOv3 repository [1]. The FasterRCNN repositories are deprecated; while Mask-RCNN and Detectron represent newer iterations, offering segmentation masks alongside bounding box predictions. This study employs the Mask-RCNN Tensorflow implementation [2] (He et al., 2017). Both were modified into biased version based on the following processes.

Sobti et al. (2021) designed synthetic biased detectors that prioritize detection based on certain visual features like size, contrast, or brightness. Unlike unbiased detectors, where detections are sampled from a log-normal probability distribution to ensure equal detection chances, biased detectors can be skewed towards certain attributes (Sobti et al., 2021). In our work, biased detectors are defined by an adjustment made to non-maximum suppression (NMS), a post-processing step commonly used in object detection tasks to remove duplicate detections, which now considers both visual attributes and confidence scores.

## 5.6 EXPERIMENTAL DESIGN

This study explores the impact of both biased datasets and bias detectors on object detection. Visual attributes including contrast, size, and brightness are computed for each dataset, enabling the categorization of images based on these attributes. The biased dataset is then created by applying grayscale conversion and blurriness. Skin tone classification for the MIAP dataset is performed using the method detailed in Section 5.1.

Four sets of experiments were conducted on each dataset, following the protocol depicted in Figure 3. The first experiment is the baseline, with normal object detectors and regular data. The second and third experiments investigated the combination of normal data and biased detector and vice versa.

---

[1] https://github.com/experiencor/keras-yolo3
[2] https://github.com/matterport/Mask_RCNN

The last experiment assessed the joint impact of biased detectors and biased datasets. Performance and fairness were assessed with the Aequitas fairness toolkit (Saleiro et al., 2018).

# 6 RESULTS

## 6.1 EXPERIMENT 1: NORMAL DATA AND NORMAL OBJECT DETECTOR

Experiment 1 is the baseline phase of our study, without biases injected into the dataset or the detector. Table 1 provides the performance metrics for the BDD dataset on both FRCNN and YOLO detectors. Despite making up 99% of the dataset, ambiguous/unknown skintones had the lowest performance of all groups, and were significant outperformed by both dark and light skin tones, which combined only comprise 1% of the dataset. For visual attributes, FRCNN consistently demonstrates superior performance to YOLO (average precision: 63% vs. 40%). Figure 4 demonstrates bias metrics for the BDD dataset when utilizing FRCNN and YOLO detectors, respectively. Notably, bias metrics remain consistent across different groups based on visual attributes such as contrast and size for both detectors.

As seen in Figure 5 and Table 2, for MIAP, unkown gender individuals exhibit by far the highest FNR and FDR and lowest TPR for both FRCNN and YOLO. Male and female individuals had far superior performance. Similarly, skin tone performance is consistent across all groups, with dark skin receiving higher AP than light skin and light skin receiving higher AP than unknown skin tones. For age, FRCNN excels with older individuals, while YOLO performs better with middle-aged individuals. Overall, YOLO performed better, particularly for the annotated attributes (age, gender, skin tones) and to a lesser degree for the visually-derived attributes (intensity, contrast).

According to Table 3, individuals with light and dark skin tones only fulfill Fairness Through Awareness (FTA) for both object detectors, while 'Unknown' skin tone upholds all fairness metrics except FTA. Both object detectors show a consistent trend across fairness metrics. Low contrast and low intensity objects pass more fairness metrics than high contrast or light objects.

For the MIAP dataset, female individuals do not satisfy SP for either detector and fail both EOP and FTA for FRCNN, while male individuals pass all fairness tests for both detectors. The unknown gender group, however, fails all fairness metrics, reflecting its high FNR and FDR. For different age groups, young and older individuals pass EO, EOP, and FTA for YOLO and EO and EOP for FRCNN. The middle age group performs slightly better, passing all metrics except FTA for FRCNN. Like gender, the unknown age group has the worst fairness performance, only passing SP for YOLO. All skin tone groups pass EO and EOP for both FRCNN and YOLO. Dark skin tone individuals additionally pass FTA for FRCNN and YOLO, and light skin tone individuals pass FTA for YOLO alone. Conversely, unknown skin tone individuals pass SP for both detectors. A summary of results can be seen in Table 4. Overall, YOLO had better fairness results than FRCNN.

## 6.2 EXPERIMENT 2: BIASED DATA AND NORMAL OBJECT DETECTOR

Figure 6 and Table 5 present the performance outcomes of both object detectors on the biased version of the BDD dataset. Both detectors exhibit considerably reduced performance, indicating their vulnerability to the data augmentations we applied. The highest AP is associated with the dark skin tone category, while the lowest AP is observed for the unknown skin tone group. Similar to the previous experiment, the different visual attribute groups demonstrate roughly equivalent performance. Across all attributes, the recall rates are low and the FNR is high. Despite its diminished performance, FRCNN outperforms YOLO in terms of AP across all attributes.

Figure 7 highlights the performance of object detectors on the biased MIAP dataset across individual visual and non-visual attributes (see Table 6 for alternate presentation). Across all attributes, YOLO performs better than FRCNN. As in Experiment 1, the lowest performance among the personal attributes for both detectors occurred with the unknown gender and unknown age categories. Male/female, young/middle aged/older, and light skin/dark skin all have fairly comparable performance for a given detector. However, YOLO performs worse on the unknown skin tone group due to the lower recall and higher FNR. For visual attributes, both detectors perform best detecting high-contrast and high-intensity objects.

Table 7 lists the pass/fail status of fairness metrics and detectors on BDD. Notably, individuals with dark and light skin tones do not meet any fairness measures compared to the reference group of unknown skin tone. All other attribute groups retain the same performance as Experiment 1.

Unknown gender individuals in MIAP fail all fairness measures for both YOLO and FRCNN. This contrasts with the female and male groups, which pass EO and EOP (males additionally pass SP). Results are similar for age. The unknown age group fails all fairness metrics, while old and young individuals pass EO and EOP for both detectors and middle-aged individuals pass EO, EOP, and SP. Unknown skin tones perform much better: they pass SP, EO, and EOP, regardless of the object detector. Among light and dark skin tone groups, FRCNN significantly outperformed YOLO. YOLO failed to meet any fairness measures, while FRCNN achieved all except SP and FTA. Regarding contrast and intensity groups, low-value groups have better results as they pass SP, EO, and EOP while high-value groups only pass EO and EOP. Interestingly, in contrast to Experiment 1, both detectors failed FTA for all groups and attributes. See Table 8 and Table 4 for a visual comparison.

### 6.3 EXPERIMENT 3: NORMAL DATA AND BIASED OBJECT DETECTOR

#### 6.3.1 CONTRAST BIAS

Table 9 presents the performance of object detectors with injected contrast bias for single attributes in BDD dataset. The highest performance is for dark skin tone individuals, closely followed by light skin tone individuals. Similar to the first experiment, the "Unknown" skin tone and visual attributes show approximately the same APs. Low-contrast individuals have higher performance than low-contrast ones. Intensity did not really make any difference in the case of APs. FRCNN displays slightly better performance, aligning with results from the baseline experiment in 1 where contrast bias was not present. According to Figure 8, FRCNN has a lower FNR than YOLO, indicating it has higher recall values on the BDD dataset. All FNR values are above 75% for both detectors. FDR and TPR values are similarly low (less than 45%). All visual attributes have roughly the same values for all metrics and detectors.

Table 10 displays the performance of object detectors on MIAP under contrast bias. FRCNN's performance ranges from 0.2 to 0.84. Among age and gender groups, the unknown groups exhibit the lowest performance; this is also reflected in the high FNR and FDR rates seen in Figure 9. The performance across different skin tones is nearly uniform. High-contrast individuals outperform their low-contrast counterparts. For intensity, dark objects achieve slightly higher APs due to their better recalls. Smaller objects fare better than larger ones. Both skin tone and visual attributes yield AP values between 0.6 and 0.7. When it comes to YOLO, performance varies between 0.2 and 0.9. The highest performance is attained by older individuals, while the lowest belongs to the age-unknown group, followed by the gender-unknown groups. YOLO demonstrates higher recall values than FRCNN, indicating a lower number of false negatives. Male and female individuals achieve the same AP, but individuals without gender exhibit significantly lower performance. Similar to FRCNN, objects with high contrast outperform those with low contrast, and those with high intensity outperform their low-intensity counterparts.

Table 11 presents fairness measures for both object detectors while contrast bias is introduced. Similar to the second experiment, light and dark skin tones did not meet fairness group measures, but both detectors passed FTA. Unknown skin tone passed all metrics except FTA for both detectors. Similar to the second experiment, groups with low value of contrast and intensity passed all metrics except FTA, while groups with high value failed both SP and FTA.

According to Table 12, YOLO passed most fairness measures for female individuals except SP, while FRCNN failed both SP and FRCNN. Male individuals outperformed all other gender groups as they passed all fairness metrics. The 'Unknown' age and 'Unknown' gender groups exhibited poor fairness performance across both object detectors, as they are not preserved due to differences between these groups and the reference group. Young and older individuals exhibited similar fairness performances, with YOLO outperforming FRCNN and both detectors failed FTA. Light and dark skin tone individuals also displayed a similar trend, with YOLO passing more fairness measures than FRCNN. For contrast, neither object detector passed the SP fairness measure for the high-contrast group and FRCNN failed FTA, too. For the low intensity group, YOLO performed better in passing all fairness measures. For high intensity, YOLO performed better as it only failed SP while FRCNN failed both SP and FTA.

### 6.3.2 INTENSITY BIAS

Table 13 illustrates the performance of object detectors under intensity bias for single attributes in BDD dataset. Results of both object detectors align with the baseline experiment in Table 1 . The highest performance is achieved by both dark and light skin tone individuals consistent with the trend observed in the absence of bias. Other visual attributes have around 0.6 APs for FRCNN and 0.3 for YOLO. Furthermore, as seen in Figure 10 , FRCNN has lower FNR values, indicating higher recall values than YOLO. FRCNN also has a higher TPR than YOLO (0.69 vs. 0.31).

Table 14 outlines performance under intensity bias for MIAP dataset. FRCNN's best performance for any attribute is for male individuals; however, it is still outperformed by YOLO, which achieves 0.90 AP for both male and female individuals. Neither FRCNN nor YOLO performs well on unknown gender individuals. Both YOLO and FRCNN see their lowest performance among unknown age individuals and much higher performance (¿0.8 AP) among middle-aged and older adult individuals. Dark, light, and unknown skin tones achieve roughly the same performance levels ( 0.60 AP) regardless of group or detector. Recall values consistently surpass precision, indicating lower confidence in detections. In Figure 11(b) , YOLO has a higher TPR and lower FNR than FRCNN for the MIAP dataset. FDR values remain consistent across all detectors and datasets.

As seen in Table 15 , when using a detector with intensity bias, light and dark skin tones did not meet fairness measures except FTA. Unknown skin tone only failed the FTA metric across both detectors. These results mirror those in Table 11 Contrast and intensity based groups, on the other hand, had higher values pass only EO and EOP and lower values pass SP, EO, and EOP.

According to Table 16 , YOLO passed most fairness measures for female individuals except SP while FRCNN failed both SP and FTA. Male individuals and middle-age individuals passed all fairness metrics. The 'Unknown' age and 'Unknown' gender groups exhibited the worst fairness performance across both object detectors. Young and older individuals as well as light and dark skin tone individuals all exhibited the same pattern of fairness performances, passing all measures except FTA for both detectors. For the high contrast group, both FRCNN and YOLO passed EO, EOP and FTA; the low contrast group passed all fairness measures. The dark intensity group also passed all fairness measures. For high intensity, YOLO performed better as it only failed SP while FRCNN failed both SP and FTA.

### 6.4 EXPERIMENT 4: BIASED DATA AND BIASED OBJECT DETECTOR

### 6.4.1 CONTRAST BIAS

Table 17 presents the performance for BDD. Both YOLO and FRCNN exhibit significantly lower performance than in baseline experiments. While the numbers differ from baseline experiment, the trend remains consistent, where dark and light skin tone individuals achieves higher APs compared to Unknown groups and groups based on visual attributes. Across all visual attributes, performance is fairly uniform for both object detectors. Overall, FRCNN yields higher numbers than YOLO, although notably smaller than baseline experiments.

Table 18 shows that for all groups, FRCNN performs at less than 0.4, with the highest performance observed in male and older individuals. Notably, many groups display low precision, indicating high values of both FP and FN. All skin tone groups exhibit similar performances, all below 0.3. YOLO performance scores are slightly better, ranging between 0.12 and 0.64. The highest scores among gender groups are on male individuals, while among age groups, older individuals take the lead, followed by middle-aged individuals. In the context of skin tone, light and dark skin tones show similar performance in both object detectors. Interestingly, the detectors exhibit divergent behaviors concerning object size. FRCNN performs better on smaller objects, whereas YOLO excels at detecting larger objects. Performance is better for larger objects, given their higher recall values.

Table 19 outlines the fairness status for both detectors in the presence of contrast biases within both object detectors regarding the BDD dataset. Both Light skin tone and Dark skin tone fail all metrics across both object detectors. Unknown gender groups pass all metrics except FTA. High value groups in both contrast and intensity only passed EO and EOP while low value in both passed also SP. Only FRCNN passes the False Tolerance Achievement (FTA) metric, while YOLO lags behind in all other metrics. None of the groups passed FTA, similar to Table 7 .

Table 20 indicates that female individuals meet EO and EOP. Male individuals meet SP, EO, and EOP. The unknown gender individual again fails all fairness measures. These results are mirrored among the age groups. Older and young individuals pass EO and EOP for both detectors; middle age groups meet SP, EO, and EOP; unknown ages fail all fairness metrics. Conversely, individuals with unknown skin tone hold SP, EO, and EOP, regardless of the object detectors used. Among light and dark skin tone groups, FRCNN significantly outperformed YOLO. YOLO failed to meet any fairness measures, while FRCNN achieved all except SP and FTA. Regarding contrast and intensity groups, low-value groups have better results as they pass SP, EO, and EOP while high value groups only pass EO and EOP. It is interesting that in comparison to Table 4 both detectors have lower performance and they failed FTA for all different groups and attributes.

### 6.4.2 INTENSITY BIAS

Table 21 reveals that FRCNN displays low performance for all groups. Both dark and light skin tone groups exhibit better performance than the unknown skin tone group, but are still significantly lower than in baseline experiments. The primary reason for FRCNN's low performance across attributes is that many ground truth boxes are not detected, leading to a high number of false negatives and subsequently, low recall values. More detail can be seen in Figure 14 . YOLO's performance similarly suffers from this issue, with AP values even lower than FRCNN's for all attributes.

Table 22 demonstrates FRCNN's performance to be less than 0.4, primarily due to low recall values. Overall, the best performance is observed in the older age group and the male group. Significant differences are evident between age and gender groups and their unknown counterparts. For visual attributes, minimal differences exist between groups, all of which exhibit low performance. YOLO's AP scores are superior to FRCNN's, even though they remain moderate. The best performance in YOLO is observed among older individuals, and the worst among unknown age groups. Concerning the skin tone attribute, the unknown group performs worse than others. The two detectors display a disparity in size attribute statistics, with FRCNN performing better on small objects and YOLO excelling at detecting large objects.

Table 23 provides fairness measures for both object detectors with the introduction of intensity bias. Similar to Table 15 , light and dark skin tones did not meet fairness measures except FTA. Unknown skin tone only failed FTA metric across both detectors. Contrast and intensity based groups have similar results as higher values passed only EO and EOP and lower values passed SP, EO, and EOP.

Table 24 presents fairness values for the MIAP dataset across both detectors. Female individuals do not meet SP fairness metrics for both detectors and FTA for FRCNN. Male and unknown gender groups got completely opposite fairness results. Among age groups, Middle individuals perform better than others, only failing the FTA metric for FRCNN, while older individuals fall behind in the FTA and SP. Similar to the "unknown" gender groups, the "Unknown" age group failed all fairness metrics. Dark and light skin tones exhibit better performance with the FRCNN detector, while YOLO fails all fairness metrics except FTA and FRCNN only fails SP. Non-reference groups for contrast and intensity display similar results for both detectors, failing FTA and SP metric. Large objects fail all metrics except FTA for YOLO and meet EO and EOP for FRCNN while small objects only failed FTA for both detectors.

## 7 DISCUSSION

### 7.1 INSIGHTS INTO OBJECT DETECTION AND BIAS

Our analyses show that the MIAP dataset, with its emphasis on individual-centric scenarios, and the BDD dataset, showcasing diverse environmental conditions, benefit from different approaches. FRCNN performed the best on the BDD dataset, potentialyl due to its segmentation of the feature space into region proposals and subsequent accumulation of detections. Conversely, for the MIAP dataset, YOLO performed best. This may be because of YOLO's ability to process images in a single step, encompassing both classification and localization; YOLO may also have benefited from higher image quality within MIAP.

## 7.2 RQ1: Implications of Biased Data for Object Detection

Framing bias introduced via grayscale conversion and blurring emerged is a potent source of performance degradation. Both transformations can naturally occur in image processing or real-world capture scenarios. This may lead to them being seen as benign, but our work shows that the biases they introduce are impactful. The ensuing decline in recall values, primarily due to elevated false negatives, has significant implications for domains like autonomous vehicles, where subtle shifts in data quality yield major impacts on object detection effectiveness. It is paramount to improve the robustness and generalization capabilities of detection methodologies. Furthermore, there is a need for tailored approaches to better detect and annotate underrepresented and unknown groups.

## 7.3 RQ2: Model Bias and Its Impacts

The introduction of bias based on visual attributes like contrast, intensity, and size led to intriguing outcomes. Impressively, FRCNN exhibited consistent performance, akin to the baseline experiment, affirming its resilience against model-induced bias. Surprisingly, the variations in impact across bias types were marginal between the two detectors. In all scenarios, performance erosion was evident for unknown groups characterized by age, gender, or skin tone, denoting compromised precision. The bias metrics unveiled a strong link between amplified data bias and the augmentation of false negative rates and the reduction in true positive rates.

## 7.4 RQ1 & RQ2: Combined Influences of Biases

The final phase of our study was an examination of the combined impact of both biased data and biased detectors on person detection outcomes. Across attributes encompassing contrast, intensity, and size, both object detectors demonstrated diminished performance relative to the baseline experiment. Additionally, both detectors exhibited worse results for unknown groups associated with age, gender, or skin tone, indicative of diminished precision in detections. Intriguingly, the type of bias injected into the model did not exert significant difference in the overall outcomes. Nevertheless, pronounced fluctuations in false negative rates and true positive rates underscored the prevalence of missed detections and low-confidence predictions versus the baseline scenario.

## 8 Conclusion

The synthesis of our study underscores the intricate interplay between bias, fairness, and object detection. These findings advocate for a holistic approach that traverses data collection, pre-processing, model conception, and evaluation to address biases effectively. The ethical dimensions of biased AI systems necessitate unwavering attention to ensure equitable and ethical deployment. The assimilation of transparent and interpretable AI techniques, coupled with collaborative endeavors, will foster the development of fair and unbiased AI systems in harmony with societal norms and values.

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

## A  APPENDIX I: ACCURACY AND FAIRNESS DEFINITIONS

### A.1  STATISTICAL METRICS

In this section we will describe statistical metrics metrics. We will give the common definition and then situate it in the object detection context.

- **True Positive (TP)**: it refers to the number of cases from a positive class that are accurately predicted as positive examples. This measure indicates the number of accurately predicted bounding boxes in object detection tasks.

- **False Positive (FP)**: it refers to the number of cases from a negative class that were incorrectly forecasted as positive. In object detection, this statistic indicates the number of predicted bounding boxes for classes that do not include any objects.

- **True Negative (TN):** it is the number of examples from a negative class that were accurately predicted as negative occurrences. The majority of object detection benchmark datasets only contain images in which the target object is present.

- **False Negative (FN):** it denotes the number of examples from a positive class that are incorrectly categorised as negative occurrences. This measure indicates the number of missed bounding boxes in an object detection situation.

- **Positive Predictive Value (PPV):** it measures how frequently predicted bounding boxes are predicted accurately in the class they belong to.

- **True Positive Rate (TPR):** it is the likelihood that, based on the actual bounding boxes in the images, a certain number of them are properly predicted.

- **False Discovery Rate (FDR):** it quantifies the frequency with which predicted bounding boxes are wrongly identified and do not include the object class given to them.

- **False Negative Rate (FNR):** it quantifies the likelihood that an actual bounding box may be overlooked by object identification models.

### A.2  STATISTICAL FAIRNESS MEASURES

In the following section, we will describe some of the most common statistical fairness measures that are derived from these measurements.

- **Statistical Parity (SP):** A classifier is deemed fair if the likelihood of a favorable outcome is the same for both privileged and unprivileged groupsDwork et al. (2012).

- **Equalized Odds (EO):** A classifier based on this metric accurately predicts positive outcomes for members of the positive and negative classes with the same probability for privileged and unprivileged groups. In conclusion, the classifier must have the same TPR and FPR for privileged and non-privileged groups Hardt et al. (2016).

- **Equal Opportunity (EOP):** A fair classifier will accurately predict the positive cases for both privileged and unprivileged groups with the same probability. That is, TPR is the same for privileged and non-privileged groups.Hardt et al. (2016).

### A.3  SIMILARITY-BASED FAIRNESS MEASURES

- **Causal Discrimination:** A classifier is considered fair based on this metric if it assigns the same categorization to any two objects with identical features X Verma & Rubin (2018); Galhotra et al. (2017).

- **Fairness Through Awareness:** A classifier is fair if cases with identical metric values receive identical predictions. The citation format is Dwork et al. (2012).

- **Fairness by ignorance:** A classifier that does not include the protected attribute in its decision-making process is regarded fair Corbett-Davies & Goel (2018).

# B  APPENDIX II: RESULTS TABLES

## B.1  EXPERIMENT 1: NORMAL DATA AND NORMAL OBJECT DETECTOR

### B.1.1  PERFORMANCE

Table 1: BDD performance analysis across single attributes either visual or demographic.

| Attribute | | FRCNN | | | | YOLO | | | |
| --- | --- | --- | --- | --- | --- | --- | --- | --- | --- |
| Attr Group | Attr Value | AP | Precision | Recall | F1 | AP | Precision | Recall | F1 |
| Skin tone | Unknown | 0.6 | 0.74 | 0.67 | 0.71 | 0.3 | 0.72 | 0.37 | 0.49 |
| Skin tone | DS | **0.94** | 0.93 | 0.94 | 0.94 | **0.96** | 0.97 | 0.97 | 0.97 |
| Skin tone | LS | 0.9 | 0.84 | 0.92 | 0.88 | 0.92 | 0.93 | 0.94 | 0.93 |
| Contrast | high | 0.59 | 0.74 | 0.66 | 0.7 | 0.34 | 0.76 | 0.39 | 0.51 |
| Contrast | low | **0.63** | 0.75 | 0.69 | 0.72 | **0.35** | 0.74 | 0.41 | 0.53 |
| Intensity | dark | **0.62** | 0.75 | 0.69 | 0.72 | 0.34 | 0.74 | 0.39 | 0.51 |
| Intensity | light | 0.62 | 0.74 | 0.69 | 0.71 | **0.4** | 0.74 | 0.46 | 0.57 |

Table 2: MIAP performance analysis across single attributes either visual or demographic

| | Attribute | FRCNN | | | | YOLO | | | |
| --- | --- | --- | --- | --- | --- | --- | --- | --- | --- |
| Attr Group | Attr Value | AP | Precision | Recall | F1 | AP | Precision | Recall | F1 |
| Gender | Predominantly Feminine | 0.78 | 0.81 | 0.81 | 0.81 | **0.93** | 0.85 | 0.97 | 0.91 |
| Gender | Predominantly Masculine | **0.84** | 0.81 | 0.87 | 0.84 | **0.93** | 0.86 | 0.96 | 0.91 |
| Gender | Unknown | 0.32 | 0.31 | 0.64 | 0.39 | 0.32 | 0.35 | 0.65 | 0.45 |
| Age | Middle | **0.82** | 0.81 | 0.85 | 0.83 | 0.93 | 0.86 | 0.96 | 0.91 |
| Age | Older | 0.81 | 0.82 | 0.83 | 0.83 | **0.96** | 0.88 | 0.98 | 0.93 |
| Age | Unknown | 0.23 | 0.26 | 0.60 | 0.31 | 0.20 | 0.28 | 0.57 | 0.31 |
| Age | Young | 0.80 | 0.81 | 0.84 | 0.82 | 0.94 | 0.85 | 0.98 | 0.91 |
| Skin tone | DS | **0.63** | 0.43 | 0.79 | 0.55 | **0.74** | 0.50 | 0.95 | 0.66 |
| Skin tone | LS | 0.59 | 0.46 | 0.72 | 0.56 | **0.74** | 0.53 | 0.92 | 0.67 |
| Skin tone | Unknown | 0.61 | 0.55 | 0.76 | 0.64 | 0.67 | 0.63 | 0.81 | 0.71 |
| Contrast | high | **0.66** | 0.61 | 0.79 | 0.68 | **0.74** | 0.67 | 0.86 | 0.75 |
| Contrast | low | 0.57 | 0.48 | 0.75 | 0.58 | 0.65 | 0.56 | 0.83 | 0.67 |
| Intensity | dark | **0.63** | 0.53 | 0.80 | 0.64 | **0.68** | 0.61 | 0.85 | 0.71 |
| Intensity | light | 0.59 | 0.51 | 0.74 | 0.60 | **0.68** | 0.59 | 0.84 | 0.69 |

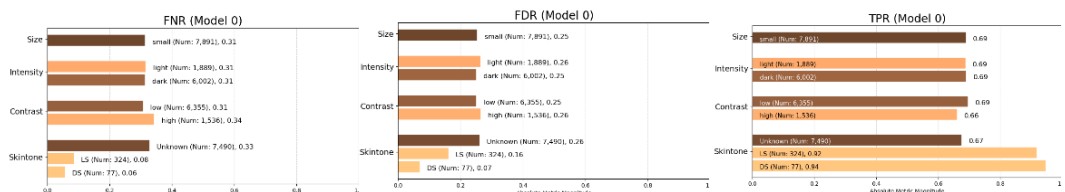

(a) BDD bias metrics FRCNN

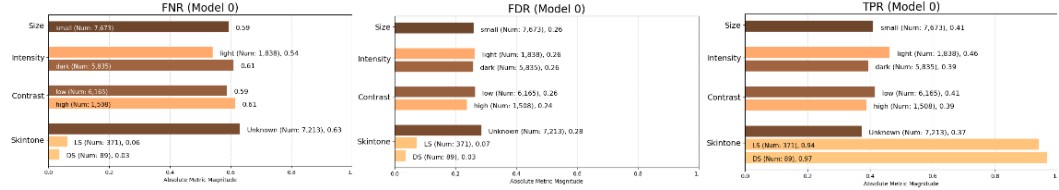

(b) BDD bias metrics YOLO

Figure 4: Experiment 1 bias metrics visualization for BDD. Photo courtesy of author.

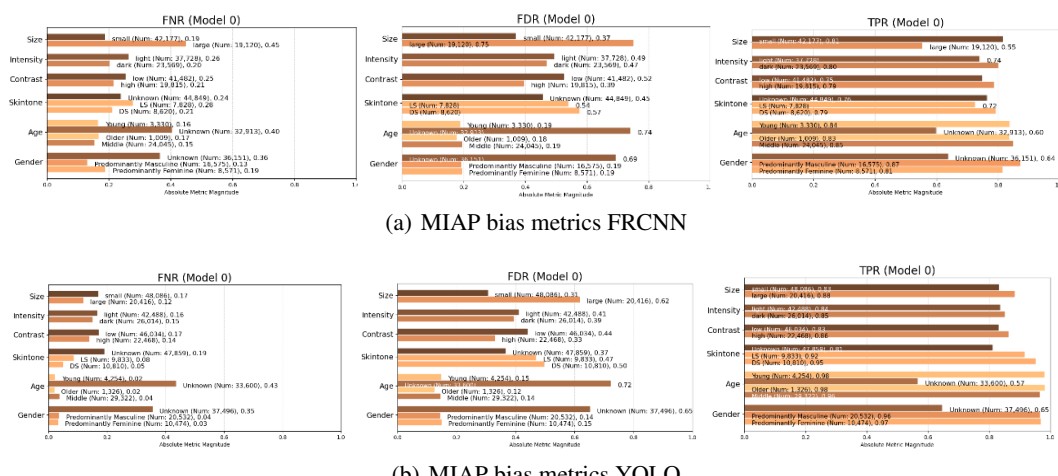

(a) MIAP bias metrics FRCNN

(b) MIAP bias metrics YOLO

Figure 5: Experiment 1 bias metrics visualization for MIAP. Photo courtesy of author.

### B.1.2 FAIRNESS

Table 3: Fairness analysis of BDD dataset in first experiment with reference group of unknown skin tone individuals with low contrast, low intensity, and small size.

| Attribute | | YOLO | | | | FRCNN | | | |
|---|---|---|---|---|---|---|---|---|---|
| Attr Group | Attr Value | SP | EO | EOP | FTA | SP | EO | EOP | FTA |
| Skin tone | DS | | | | ✓ | | | | ✓ |
| Skin tone | LS | | | | ✓ | | | | ✓ |
| Skin tone | Unknown | ✓ | ✓ | ✓ | | ✓ | ✓ | ✓ | |
| Contrast | high | | ✓ | ✓ | | | ✓ | ✓ | |
| Contrast | low | ✓ | ✓ | ✓ | | ✓ | ✓ | ✓ | |
| Intensity | dark | ✓ | ✓ | ✓ | | ✓ | ✓ | ✓ | |
| Intensity | light | | ✓ | ✓ | | | ✓ | ✓ | height |

Table 4: Fairness analysis of MIAP dataset in first experiment with reference group of Middle-aged male individuals with unknown skin tone, low-contrast, low intensity, and small size.

| Attribute | | FRCNN | | | | YOLO | | | |
|---|---|---|---|---|---|---|---|---|---|
| Attr Group | Attr Value | SP | EO | EOP | FTA | SP | EO | EOP | FTA |
| Gender | Predominantly Feminine | | ✓ | | | | ✓ | ✓ | ✓ |
| Gender | Predominantly Masculine | ✓ | ✓ | ✓ | ✓ | ✓ | ✓ | ✓ | ✓ |
| Gender | Unknown | | | | | | | | |
| Age | Middle | ✓ | ✓ | ✓ | | ✓ | ✓ | ✓ | ✓ |
| Age | Older | | ✓ | ✓ | | | ✓ | ✓ | ✓ |
| Age | Unknown | | | | | ✓ | | | |
| Age | Young | | ✓ | ✓ | | | ✓ | ✓ | ✓ |
| Skin tone | DS | | ✓ | ✓ | ✓ | | ✓ | ✓ | ✓ |
| Skin tone | LS | | ✓ | ✓ | | | ✓ | ✓ | ✓ |
| Skin tone | Unknown | ✓ | ✓ | ✓ | | ✓ | ✓ | ✓ | |
| Contrast | high | | ✓ | ✓ | | | ✓ | ✓ | ✓ |
| Contrast | low | ✓ | ✓ | ✓ | | ✓ | ✓ | ✓ | ✓ |
| Intensity | dark | ✓ | ✓ | ✓ | ✓ | ✓ | ✓ | ✓ | ✓ |
| Intensity | light | | ✓ | ✓ | | | ✓ | ✓ | ✓ |

## B.2 EXPERIMENT 2: BIASED DATA AND NORMAL OBJECT DETECTOR

### B.2.1 PERFORMANCE

Table 5: BDD performance analysis across single attributes either visual or demographic

| Attribute | | FRCNN | | | | YOLO | | | |
|---|---|---|---|---|---|---|---|---|---|
| Attr Group | Attr Value | AP | Precision | Recall | F1 | AP | Precision | Recall | F1 |
| Skin tone | Unknown | 0.22 | 0.71 | 0.25 | 0.35 | 0.10 | 0.69 | 0.12 | 0.17 |
| Skin tone | DS | **0.45** | 0.91 | 0.45 | 0.60 | 0.43 | 0.91 | 0.44 | 0.59 |
| Skin tone | LS | 0.38 | 0.82 | 0.39 | 0.53 | **0.44** | 0.95 | 0.44 | 0.60 |
| Contrast | high | **0.23** | 0.72 | 0.26 | 0.37 | **0.12** | 0.76 | 0.14 | 0.21 |
| Contrast | low | 0.22 | 0.71 | 0.25 | 0.36 | 0.11 | 0.71 | 0.14 | 0.19 |
| Intensity | dark | **0.23** | 0.72 | 0.26 | 0.37 | 0.11 | 0.72 | 0.14 | 0.20 |
| Intensity | light | 0.22 | 0.70 | 0.24 | 0.33 | **0.12** | 0.73 | 0.14 | 0.20 |
| Size | small | **0.23** | 0.71 | 0.25 | 0.36 | **0.11** | 0.72 | 0.14 | 0.20 |

Table 6: MIAP performance analysis across single attributes either visual or demographic

| | Attribute | FRCNN | | | | YOLO | | | |
|---|---|---|---|---|---|---|---|---|---|
| Attr Group | Attr Value | AP | Precision | Recall | F1 | AP | Precision | Recall | F1 |
| Gender | Predominantly Feminine | 0.33 | 0.80 | 0.35 | 0.49 | **0.57** | 0.89 | 0.59 | 0.71 |
| Gender | Predominantly Masculine | **0.38** | 0.79 | 0.40 | 0.53 | 0.55 | 0.89 | 0.57 | 0.69 |
| Gender | Unknown | 0.12 | 0.31 | 0.23 | 0.14 | 0.14 | 0.40 | 0.26 | 0.21 |
| Age | Middle | 0.36 | 0.79 | 0.38 | 0.51 | 0.55 | 0.89 | 0.57 | 0.69 |
| Age | Older | **0.38** | 0.80 | 0.39 | 0.53 | **0.65** | 0.90 | 0.67 | 0.77 |
| Age | Unknown | 0.08 | 0.26 | 0.21 | 0.11 | 0.08 | 0.31 | 0.21 | 0.13 |
| Age | Young | 0.35 | 0.83 | 0.36 | 0.51 | 0.57 | 0.87 | 0.60 | 0.71 |
| Skin tone | DS | **0.27** | 0.43 | 0.36 | 0.31 | 0.52 | 0.59 | 0.66 | 0.63 |
| Skin tone | LS | 0.27 | 0.48 | 0.35 | 0.33 | **0.53** | 0.63 | 0.65 | 0.64 |
| Skin tone | Unknown | 0.24 | 0.53 | 0.30 | 0.32 | 0.32 | 0.69 | 0.37 | 0.48 |
| Contrast | high | **0.28** | 0.60 | 0.33 | 0.40 | **0.40** | 0.73 | 0.47 | 0.57 |
| Contrast | low | 0.22 | 0.47 | 0.30 | 0.28 | 0.34 | 0.63 | 0.42 | 0.50 |
| Intensity | dark | 0.24 | 0.51 | 0.31 | 0.32 | 0.32 | 0.65 | 0.39 | 0.49 |
| Intensity | light | **0.25** | 0.51 | 0.31 | 0.32 | **0.38** | 0.67 | 0.46 | 0.54 |
| Size | large | 0.18 | 0.29 | 0.30 | 0.17 | **0.49** | 0.51 | 0.64 | 0.57 |
| Size | small | **0.27** | 0.62 | 0.31 | 0.39 | 0.35 | 0.74 | 0.39 | 0.51 |

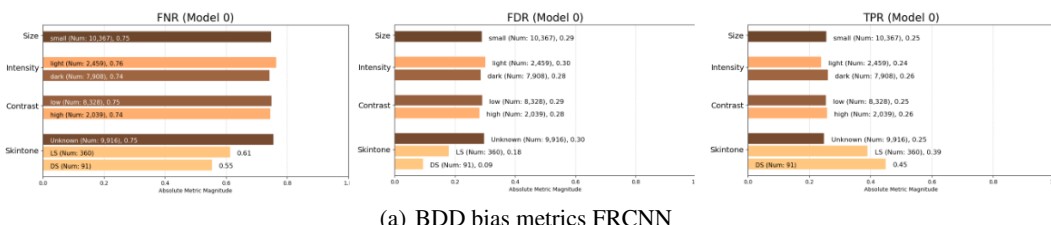

(a) BDD bias metrics FRCNN

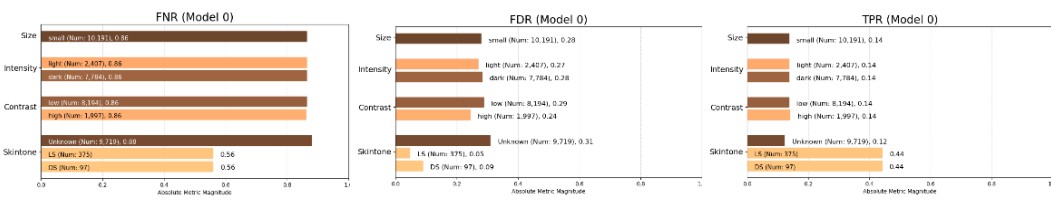

(b) BDD bias metrics YOLO

Figure 6: Experiment 2 bias metrics visualization for BDD. Photo courtesy of author.

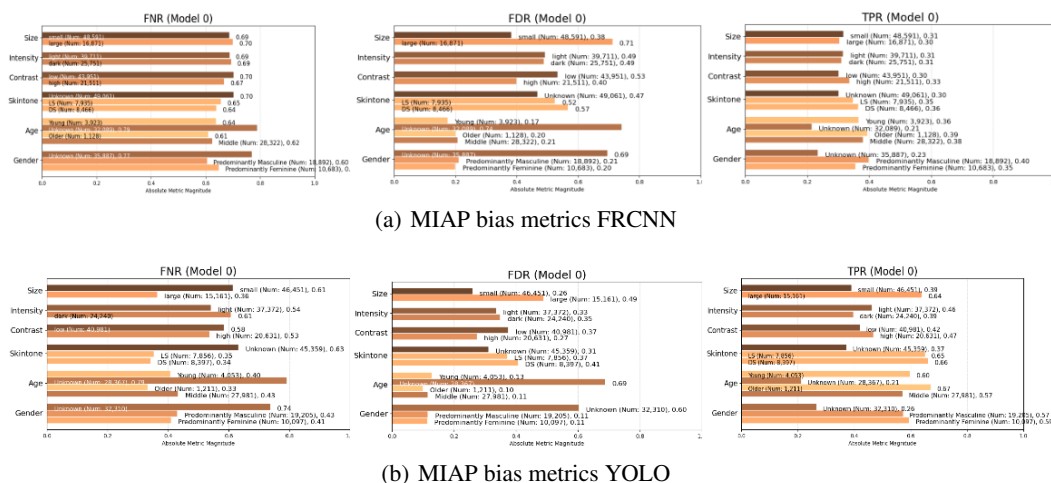

(a) MIAP bias metrics FRCNN

(b) MIAP bias metrics YOLO

Figure 7: Experiment 2 bias metrics visualization for MIAP. Photo courtesy of author.

## B.2.2 FAIRNESS

Table 7: Fairness analysis of BDD dataset in second experiment with reference group of unknown skin tone individuals with low contrast, low intensity, and small size.

| Attribute | | YOLO | | | | FRCNN | | | |
|---|---|---|---|---|---|---|---|---|---|
| Attr Group | Attr Value | SP | EO | EOP | FTA | SP | EO | EOP | FTA |
| Skin tone | DS | | | | | | | | |
| Skin tone | LS | | | | | | | | |
| Skin tone | Unknown | ✓ | ✓ | ✓ | | ✓ | ✓ | ✓ | |
| Contrast | high | | ✓ | ✓ | | | ✓ | ✓ | |
| Contrast | low | ✓ | ✓ | ✓ | | ✓ | ✓ | ✓ | |
| Intensity | dark | ✓ | ✓ | ✓ | | ✓ | ✓ | ✓ | |
| Intensity | light | | ✓ | ✓ | | | ✓ | ✓ | |

Table 8: Fairness analysis of MIAP dataset in the second experiment with a reference group of Middle-aged male individuals with unknown skin tone, low-contrast, low intensity, and small size.

| Attribute | | FRCNN | | | | YOLO | | | |
|---|---|---|---|---|---|---|---|---|---|
| Attr Group | Attr Value | SP | EO | EOP | FTA | SP | EO | EOP | FTA |
| Gender | Predominantly Feminine | | ✓ | ✓ | | | ✓ | ✓ | |
| Gender | Predominantly Masculine | ✓ | ✓ | ✓ | | ✓ | ✓ | ✓ | |
| Gender | Unknown | | | | | | | | |
| Age | Middle | ✓ | ✓ | ✓ | | ✓ | ✓ | ✓ | |
| Age | Older | | ✓ | ✓ | | | ✓ | ✓ | |
| Age | Unknown | | | | | | | | |
| Age | Young | | ✓ | ✓ | | | ✓ | ✓ | |
| Skin tone | DS | | ✓ | ✓ | | | | | |
| Skin tone | LS | | ✓ | ✓ | | | | | |
| Skin tone | Unknown | ✓ | ✓ | ✓ | | ✓ | ✓ | ✓ | |
| Contrast | high | | ✓ | ✓ | | | ✓ | ✓ | |
| Contrast | low | ✓ | ✓ | ✓ | | ✓ | ✓ | ✓ | |
| Intensity | dark | ✓ | ✓ | ✓ | | ✓ | ✓ | ✓ | |
| Intensity | light | | ✓ | ✓ | | | ✓ | ✓ | |

### B.3 EXPERIMENT 3: NORMAL DATA AND BIASED OBJECT DETECTOR

#### B.3.1 CONTRAST BIAS PERFORMANCE

Table 9: BDD performance analysis across single attributes either visual or demographic based on contrast bias

| Attribute | | FRCNN | | | | YOLO | | | |
|---|---|---|---|---|---|---|---|---|---|
| Attr Group | Attr Value | AP | Precision | Recall | F1 | AP | Precision | Recall | F1 |
| Skin tone | Unknown | **0.60** | 0.74 | 0.67 | 0.71 | **0.30** | 0.72 | 0.37 | 0.49 |
| Skin tone | DS | **0.94** | 0.93 | 0.94 | 0.94 | **0.96** | 0.93 | 0.96 | 0.95 |
| Skin tone | LS | 0.90 | 0.84 | 0.92 | 0.88 | 0.91 | 0.91 | 0.94 | 0.93 |
| Contrast | high | 0.59 | 0.74 | 0.66 | 0.70 | 0.34 | 0.77 | 0.39 | 0.51 |
| Contrast | low | **0.63** | 0.75 | 0.69 | 0.72 | 0.34 | 0.73 | 0.41 | 0.53 |
| Intensity | dark | 0.62 | 0.75 | 0.69 | 0.72 | 0.33 | 0.74 | 0.39 | 0.51 |
| Intensity | light | 0.62 | 0.74 | 0.69 | 0.71 | 0.38 | 0.73 | 0.46 | 0.56 |
| Size | small | 0.62 | 0.75 | 0.69 | 0.72 | 0.34 | 0.74 | 0.41 | 0.53 |

Table 10: MIAP performance analysis across single attributes either visual or demographic based on contrast bias

| Attribute | | FRCNN | | | | YOLO | | | |
|---|---|---|---|---|---|---|---|---|---|
| Attr Group | Attr Value | AP | Precision | Recall | F1 | AP | Precision | Recall | F1 |
| Gender | Predominantly Feminine | 0.78 | 0.81 | 0.81 | 0.81 | **0.90** | 0.85 | 0.97 | 0.90 |
| Gender | Predominantly Masculine | **0.84** | 0.81 | 0.87 | 0.84 | **0.90** | 0.85 | 0.96 | 0.90 |
| Gender | Unknown | **0.32** | 0.31 | 0.64 | 0.39 | 0.27 | 0.35 | 0.65 | 0.45 |
| Age | Middle | **0.82** | 0.81 | 0.85 | 0.83 | 0.90 | 0.85 | 0.96 | 0.90 |
| Age | Older | 0.81 | 0.82 | 0.83 | 0.83 | **0.92** | 0.86 | 0.98 | 0.92 |
| Age | Unknown | **0.23** | 0.26 | 0.60 | 0.31 | **0.17** | 0.28 | 0.57 | 0.31 |
| Age | Young | 0.80 | 0.81 | 0.84 | 0.82 | 0.90 | 0.85 | 0.98 | 0.91 |
| Skin tone | DS | **0.63** | 0.42 | 0.79 | 0.55 | 0.61 | 0.50 | 0.95 | 0.66 |
| Skin tone | LS | **0.59** | 0.46 | 0.72 | 0.56 | 0.60 | 0.53 | 0.92 | 0.67 |
| Skin tone | Unknown | 0.61 | 0.54 | 0.76 | 0.64 | 0.61 | 0.63 | 0.81 | 0.71 |
| Contrast | high | **0.66** | 0.60 | 0.79 | 0.68 | **0.67** | 0.67 | 0.86 | 0.75 |
| Contrast | low | 0.57 | 0.47 | 0.75 | 0.58 | 0.56 | 0.56 | 0.83 | 0.67 |
| Intensity | dark | **0.63** | 0.53 | 0.80 | 0.64 | **0.62** | 0.60 | 0.85 | 0.71 |
| Intensity | light | 0.59 | 0.51 | 0.74 | 0.60 | 0.59 | 0.58 | 0.83 | 0.69 |
| Size | large | 0.35 | 0.25 | 0.55 | 0.28 | 0.41 | 0.38 | 0.88 | 0.53 |
| Size | small | **0.71** | 0.63 | 0.81 | 0.71 | **0.68** | 0.69 | 0.83 | 0.75 |

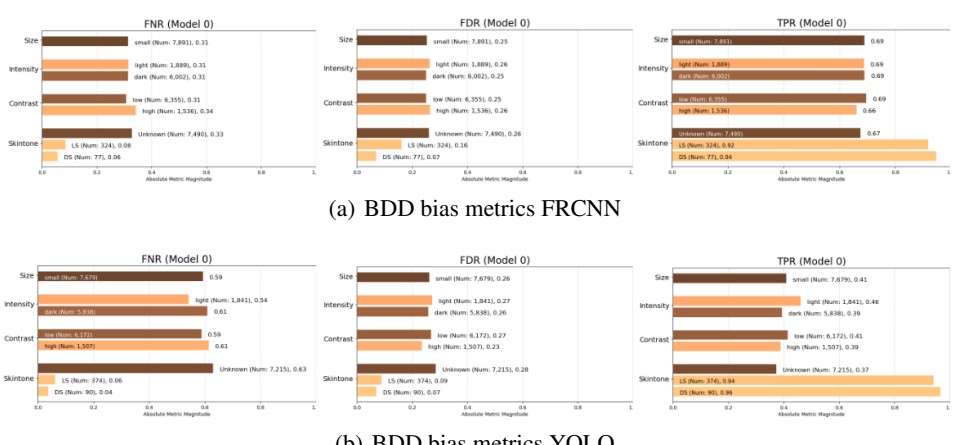

(a) BDD bias metrics FRCNN

(b) BDD bias metrics YOLO

Figure 8: Experiment 3 bias metrics visualization for BDD while injecting contrast biases into the object detector. Photo courtesy of author.

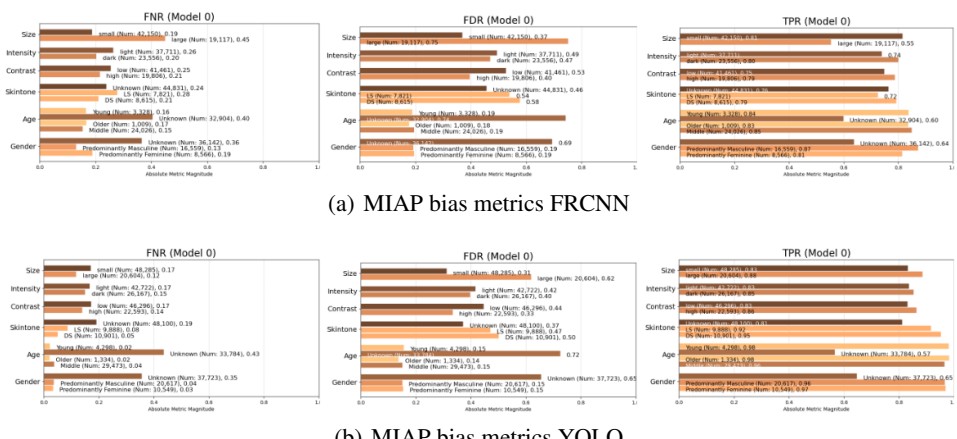

(a) MIAP bias metrics FRCNN

(b) MIAP bias metrics YOLO

Figure 9: Experiment 3 bias metrics visualization for MIAP while injecting contrast biases into the object detector. Photo courtesy of author.

### B.3.2 CONTRAST BIAS FAIRNESS

Table 11: Fairness analysis of BDD dataset in the third experiment with reference group of unknown skin tone individuals with low contrast, low intensity, and small size. Here biased detector is created based on the contrast attribute.

| Attribute | | YOLO | | | | FRCNN | | | |
|---|---|---|---|---|---|---|---|---|---|
| Attr Group | Attr Value | SP | EO | EOP | FTA | SP | EO | EOP | FTA |
| Skin tone | DS | | | | ✓ | | | | ✓ |
| Skin tone | LS | | | | ✓ | | | | ✓ |
| Skin tone | Unknown | ✓ | ✓ | ✓ | | ✓ | ✓ | ✓ | |
| Contrast | high | | ✓ | ✓ | | | ✓ | ✓ | |
| Contrast | low | ✓ | ✓ | ✓ | | ✓ | ✓ | ✓ | |
| Intensity | dark | ✓ | ✓ | ✓ | | ✓ | ✓ | ✓ | |
| Intensity | light | | ✓ | ✓ | | | ✓ | ✓ | |

Table 12: Fairness analysis of MIAP dataset in third experiment with reference group of Middle-aged male individuals with unknown skin tone, low-contrast, low intensity, and small size. Here biased detector is created based on contrast attribute.

| Attribute | | FRCNN | | | | YOLO | | | |
|---|---|---|---|---|---|---|---|---|---|
| Attr Group | Attr Value | SP | EO | EOP | FTA | SP | EO | EOP | FTA |
| Gender | Predominantly Feminine | | ✓ | ✓ | | | ✓ | ✓ | ✓ |
| Gender | Predominantly Masculine | ✓ | ✓ | ✓ | ✓ | ✓ | ✓ | ✓ | ✓ |
| Gender | Unknown | | | | | | | | |
| Age | Middle | ✓ | ✓ | ✓ | | ✓ | ✓ | ✓ | ✓ |
| Age | Older | | ✓ | ✓ | | ✓ | ✓ | ✓ | ✓ |
| Age | Unknown | | | | | ✓ | | | |
| Age | Young | | ✓ | ✓ | | ✓ | ✓ | ✓ | ✓ |
| Skin tone | DS | | ✓ | ✓ | ✓ | | ✓ | ✓ | ✓ |
| Skin tone | LS | | ✓ | ✓ | | | ✓ | ✓ | ✓ |
| Skin tone | Unknown | ✓ | ✓ | ✓ | | ✓ | ✓ | ✓ | |
| Contrast | high | | ✓ | ✓ | | | ✓ | ✓ | ✓ |
| Contrast | low | ✓ | ✓ | ✓ | | ✓ | ✓ | ✓ | ✓ |
| Intensity | dark | ✓ | ✓ | ✓ | ✓ | ✓ | ✓ | ✓ | ✓ |
| Intensity | light | | ✓ | ✓ | | | ✓ | ✓ | ✓ |

### B.3.3 INTENSITY BIAS PERFORMANCE

Table 13: BDD performance analysis across single attributes either visual or demographic based on intensity bias.

| Attribute | | FRCNN | | | | YOLO | | | |
|---|---|---|---|---|---|---|---|---|---|
| Attr Group | Attr Value | AP | Precision | Recall | F1 | AP | Precision | Recall | F1 |
| Skin tone | Unknown | 0.60 | 0.74 | 0.67 | 0.71 | 0.30 | 0.72 | 0.37 | 0.49 |
| Skin tone | DS | **0.94** | 0.93 | 0.94 | 0.94 | **0.96** | 0.95 | 0.96 | 0.96 |
| Skin tone | LS | 0.90 | 0.84 | 0.92 | 0.88 | 0.91 | 0.92 | 0.94 | 0.93 |
| Contrast | high | 0.59 | 0.74 | 0.66 | 0.70 | 0.34 | 0.76 | 0.39 | 0.51 |
| Contrast | low | **0.63** | 0.75 | 0.69 | 0.72 | **0.35** | 0.74 | 0.41 | 0.53 |
| Intensity | dark | 0.62 | 0.75 | 0.69 | 0.72 | 0.33 | 0.74 | 0.39 | 0.51 |
| Intensity | light | 0.62 | 0.74 | 0.69 | 0.71 | **0.39** | 0.74 | 0.46 | 0.57 |
| Size | small | 0.62 | 0.75 | 0.69 | 0.72 | 0.34 | 0.74 | 0.41 | 0.53 |

Table 14: MIAP performance analysis across single attributes either visual or demographic based on intensity bias

| Attribute | | FRCNN | | | | YOLO | | | |
|---|---|---|---|---|---|---|---|---|---|
| Attr Group | Attr Value | AP | Precision | Recall | F1 | AP | Precision | Recall | F1 |
| Gender | Predominantly Feminine | 0.78 | 0.81 | 0.81 | 0.81 | **0.90** | 0.85 | 0.97 | 0.90 |
| Gender | Predominantly Masculine | **0.84** | 0.81 | 0.87 | 0.84 | **0.90** | 0.85 | 0.96 | 0.90 |
| Gender | Unknown | **0.32** | 0.31 | 0.64 | 0.39 | 0.27 | 0.35 | 0.65 | 0.45 |
| Age | Middle | **0.82** | 0.81 | 0.85 | 0.83 | 0.90 | 0.85 | 0.96 | 0.90 |
| Age | Older | 0.81 | 0.82 | 0.83 | 0.83 | **0.93** | 0.87 | 0.98 | 0.92 |
| Age | Unknown | **0.23** | 0.26 | 0.60 | 0.31 | 0.18 | 0.28 | 0.57 | 0.31 |
| Age | Young | 0.80 | 0.81 | 0.84 | 0.82 | 0.90 | 0.85 | 0.98 | 0.91 |
| Skin tone | DS | **0.63** | 0.42 | 0.79 | 0.55 | **0.62** | 0.50 | 0.95 | 0.66 |
| Skin tone | LS | **0.59** | 0.46 | 0.72 | 0.56 | 0.61 | 0.53 | 0.92 | 0.67 |
| Skin tone | Unknown | 0.61 | 0.54 | 0.76 | 0.64 | **0.62** | 0.63 | 0.81 | 0.71 |
| Contrast | high | **0.66** | 0.60 | 0.79 | 0.68 | **0.68** | 0.67 | 0.86 | 0.75 |
| Contrast | low | 0.57 | 0.47 | 0.75 | 0.58 | 0.58 | 0.56 | 0.83 | 0.67 |
| Intensity | dark | **0.63** | 0.53 | 0.80 | 0.64 | **0.62** | 0.60 | 0.85 | 0.71 |
| Intensity | light | 0.59 | 0.51 | 0.74 | 0.60 | 0.61 | 0.59 | 0.84 | 0.69 |
| Size | large | 0.35 | 0.25 | 0.55 | 0.28 | 0.45 | 0.38 | 0.88 | 0.53 |
| Size | small | **0.71** | 0.63 | 0.81 | 0.71 | **0.69** | 0.69 | 0.83 | 0.75 |

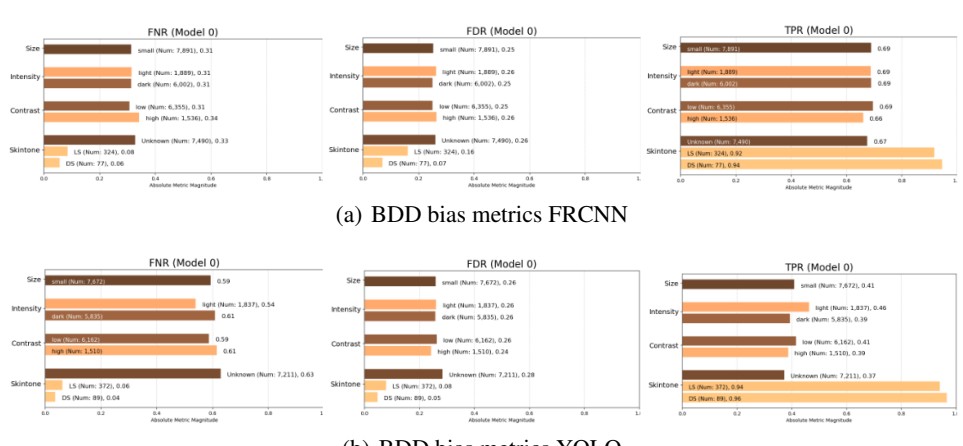

(a) BDD bias metrics FRCNN

(b) BDD bias metrics YOLO

Figure 10: Experiment 3 bias metrics visualization for BDD while injecting intensity biases into the object detector. Photo courtesy of author.

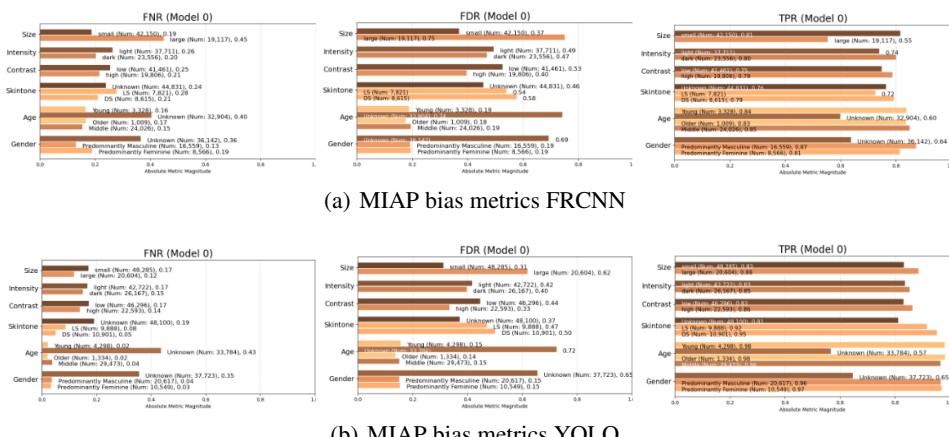

(a) MIAP bias metrics FRCNN

(b) MIAP bias metrics YOLO

Figure 11: Experiment 3 bias metrics visualization for MIAP while injecting intensity biases into the object detector. Photo courtesy of author.

### B.3.4 INTENSITY BIAS FAIRNESS

Table 15: Fairness analysis of BDD dataset in third experiment with reference group of unknown skin tone individuals with low contrast, low intensity, and small size.Here biased detector is created based on intensity attribute.

| Attribute | | YOLO | | | | FRCNN | | | |
|---|---|---|---|---|---|---|---|---|---|
| Attr Group | Attr Value | SP | EO | EOP | FTA | SP | EO | EOP | FTA |
| Skin tone | DS | | | | ✓ | | | | ✓ |
| Skin tone | LS | | | | ✓ | | | | ✓ |
| Skin tone | Unknown | ✓ | ✓ | ✓ | | ✓ | ✓ | ✓ | |
| Contrast | high | | ✓ | ✓ | | | ✓ | ✓ | |
| Contrast | low | ✓ | ✓ | ✓ | | ✓ | ✓ | ✓ | |
| Intensity | dark | ✓ | ✓ | ✓ | | ✓ | ✓ | ✓ | |
| Intensity | light | | ✓ | ✓ | | | ✓ | ✓ | |

Table 16: Fairness analysis of MIAP dataset in third experiment with reference group of Middle-aged male individuals with unknown skin tone, low-contrast, low intensity, and small size. Here biased detector is created based on intensity attribute.

| Attribute | | FRCNN | | | | YOLO | | | |
|---|---|---|---|---|---|---|---|---|---|
| Attr Group | Attr Value | SP | EO | EOP | FTA | SP | EO | EOP | FTA |
| Gender | Predominantly Feminine | | ✓ | ✓ | | | ✓ | ✓ | ✓ |
| Gender | Predominantly Masculine | ✓ | ✓ | ✓ | ✓ | ✓ | ✓ | ✓ | ✓ |
| Gender | Unknown | | | | | | | | ✓ |
| Age | Middle | ✓ | ✓ | ✓ | ✓ | ✓ | ✓ | ✓ | ✓ |
| Age | Older | | ✓ | ✓ | ✓ | | ✓ | ✓ | ✓ |
| Age | Unknown | | | | | ✓ | | | |
| Age | Young | | ✓ | ✓ | ✓ | | ✓ | ✓ | ✓ |
| Skin tone | DS | | ✓ | ✓ | ✓ | | ✓ | ✓ | ✓ |
| Skin tone | LS | | ✓ | ✓ | ✓ | | ✓ | ✓ | ✓ |
| Skin tone | Unknown | ✓ | ✓ | ✓ | ✓ | ✓ | ✓ | ✓ | |
| Contrast | high | | ✓ | ✓ | ✓ | | ✓ | ✓ | ✓ |
| Contrast | low | ✓ | ✓ | ✓ | ✓ | ✓ | ✓ | ✓ | ✓ |
| Intensity | dark | ✓ | ✓ | ✓ | ✓ | ✓ | ✓ | ✓ | ✓ |
| Intensity | light | | ✓ | ✓ | | | ✓ | ✓ | ✓ |

## B.4 Experiment 4: Biased Data and Biased Object Detector

### B.4.1 Contrast Bias Performance

Table 17: BDD performance analysis across single attributes either visual or demographic based on contrast bias

| Attribute | | FRCNN | | | | YOLO | | | |
| Attr Group | Attr Value | AP | Precision | Recall | F1 | AP | Precision | Recall | F1 |
| --- | --- | --- | --- | --- | --- | --- | --- | --- | --- |
| Skin tone | Unknown | 0.22 | 0.71 | 0.25 | 0.35 | 0.10 | 0.69 | 0.12 | 0.17 |
| Skin tone | DS | **0.45** | 0.91 | 0.45 | 0.60 | **0.43** | 0.91 | 0.44 | 0.59 |
| Skin tone | LS | **0.38** | 0.82 | 0.39 | 0.53 | **0.43** | 0.95 | 0.44 | 0.60 |
| Contrast | high | **0.23** | 0.72 | 0.26 | 0.37 | **0.12** | 0.76 | 0.14 | 0.21 |
| Contrast | low | 0.22 | 0.71 | 0.25 | 0.36 | 0.11 | 0.71 | 0.14 | 0.20 |
| Intensity | dark | **0.23** | 0.72 | 0.26 | 0.37 | 0.11 | 0.72 | 0.14 | 0.20 |
| Intensity | light | 0.22 | 0.70 | 0.24 | 0.33 | **0.12** | 0.73 | 0.14 | 0.20 |
| Size | small | 0.23 | 0.71 | 0.25 | 0.36 | 0.11 | 0.72 | 0.14 | 0.20 |

Table 18: MIAP performance analysis across single attributes either visual or demographic based on contrast bias.

| Attribute | | FRCNN | | | | YOLO | | | |
| Attr Group | Attr Value | AP | Precision | Recall | F1 | AP | Precision | Recall | F1 |
| --- | --- | --- | --- | --- | --- | --- | --- | --- | --- |
| Gender | Predominantly Feminine | 0.33 | 0.8 | 0.35 | 0.49 | **0.55** | 0.88 | 0.59 | 0.71 |
| Gender | Predominantly Masculine | **0.38** | 0.79 | 0.4 | 0.53 | 0.54 | 0.88 | 0.57 | 0.69 |
| Gender | Unknown | **0.12** | 0.31 | 0.23 | 0.14 | **0.12** | 0.4 | 0.27 | 0.21 |
| Age | Middle | 0.36 | 0.79 | 0.38 | 0.51 | 0.53 | 0.88 | 0.57 | 0.69 |
| Age | Older | **0.38** | 0.8 | 0.39 | 0.53 | **0.64** | 0.89 | 0.67 | 0.76 |
| Age | Unknown | **0.08** | 0.26 | 0.21 | 0.11 | **0.07** | 0.31 | 0.21 | 0.13 |
| Age | Young | 0.35 | 0.83 | 0.36 | 0.51 | 0.56 | 0.87 | 0.6 | 0.71 |
| Skin tone | DS | **0.27** | 0.43 | 0.36 | 0.31 | **0.45** | 0.59 | 0.66 | 0.62 |
| Skin tone | LS | **0.27** | 0.48 | 0.35 | 0.33 | **0.45** | 0.63 | 0.65 | 0.64 |
| Skin tone | Unknown | **0.24** | 0.53 | 0.3 | 0.32 | **0.29** | 0.69 | 0.37 | 0.48 |
| Contrast | high | **0.28** | 0.6 | 0.33 | 0.4 | **0.38** | 0.72 | 0.47 | 0.57 |
| Contrast | low | 0.22 | 0.47 | 0.3 | 0.28 | 0.3 | 0.62 | 0.42 | 0.5 |
| Intensity | dark | 0.24 | 0.51 | 0.31 | 0.32 | 0.3 | 0.65 | 0.4 | 0.49 |
| Intensity | light | **0.25** | 0.51 | 0.31 | 0.32 | **0.34** | 0.66 | 0.46 | 0.54 |
| Size | large | 0.18 | 0.29 | 0.3 | 0.17 | **0.36** | 0.51 | 0.64 | 0.57 |
| Size | small | **0.27** | 0.62 | 0.31 | 0.39 | 0.33 | 0.74 | 0.39 | 0.51 |

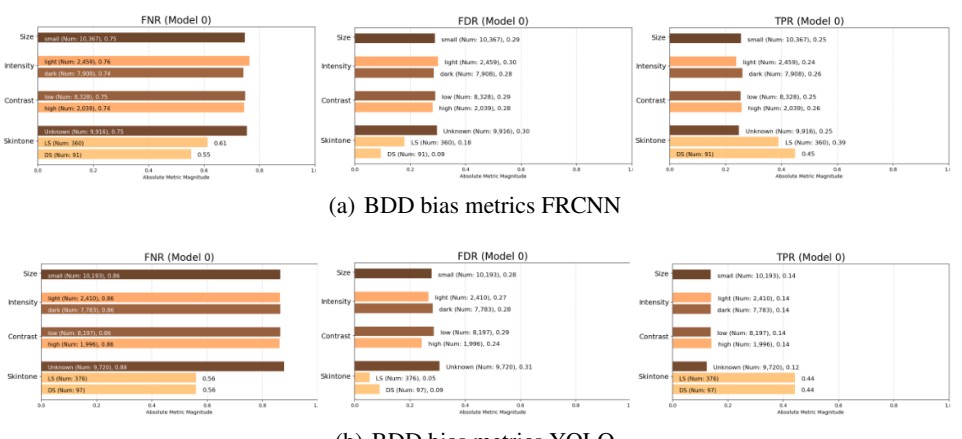

(a) BDD bias metrics FRCNN

(b) BDD bias metrics YOLO

Figure 12: Experiment 4 bias metrics visualization for BDD while injecting contrast biases into the object detector. Photo courtesy of author.

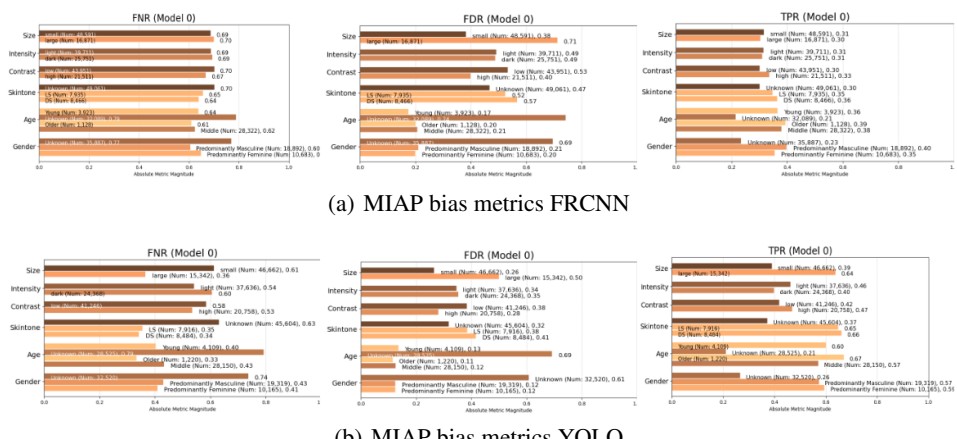

(a) MIAP bias metrics FRCNN

(b) MIAP bias metrics YOLO

Figure 13: Experiment 4 bias metrics visualization for MIAP while injecting contrast biases into the object detector. Photo courtesy of author.

### B.4.2 CONTRAST BIAS FAIRNESS

Table 19: Fairness analysis of BDD dataset in fourth experiment with reference group of unknown skin tone individuals with low contrast, low intensity, and small size. Contrast bias is injected to the detector.

| Attribute | | YOLO | | | | FRCNN | | | |
|---|---|---|---|---|---|---|---|---|---|
| Attr Group | Attr Value | SP | EO | EOP | FTA | SP | EO | EOP | FTA |
| Skin tone | DS | | | | | | | | |
| Skin tone | LS | | | | | | | | |
| Skin tone | Unknown | ✓ | ✓ | ✓ | | ✓ | ✓ | ✓ | |
| Contrast | high | | ✓ | ✓ | | | ✓ | ✓ | |
| Contrast | low | ✓ | ✓ | ✓ | | ✓ | ✓ | ✓ | |
| Intensity | dark | ✓ | ✓ | ✓ | | ✓ | ✓ | ✓ | |
| Intensity | light | | ✓ | ✓ | | | ✓ | ✓ | |

Table 20: Fairness analysis of MIAP dataset with reference group of Middle-aged male individuals with unknown skin tone, low-contrast, low intensity, and small size. Contrast bias is injected to the detector.

| Attribute | | FRCNN | | | | YOLO | | | |
|---|---|---|---|---|---|---|---|---|---|
| Attr Group | Attr Value | SP | EO | EOP | FTA | SP | EO | EOP | FTA |
| Gender | Predominantly Feminine | | ✓ | ✓ | | | ✓ | ✓ | |
| Gender | Predominantly Masculine | ✓ | ✓ | ✓ | | ✓ | ✓ | ✓ | |
| Gender | Unknown | | | | | | | | |
| Age | Middle | ✓ | ✓ | ✓ | | ✓ | ✓ | ✓ | |
| Age | Older | | ✓ | ✓ | | | ✓ | ✓ | |
| Age | Unknown | | | | | | | | |
| Age | Young | | ✓ | ✓ | | | ✓ | ✓ | |
| Skin tone | DS | | ✓ | ✓ | | | | | |
| Skin tone | LS | | ✓ | ✓ | | | | | |
| Skin tone | Unknown | ✓ | ✓ | ✓ | | ✓ | ✓ | ✓ | |
| Contrast | high | | ✓ | ✓ | | | ✓ | ✓ | |
| Contrast | low | ✓ | ✓ | ✓ | | ✓ | ✓ | ✓ | |
| Intensity | dark | ✓ | ✓ | ✓ | | ✓ | ✓ | ✓ | |
| Intensity | light | | ✓ | ✓ | | | ✓ | ✓ | |

### B.4.3 INTENSITY BIAS PERFORMANCE

Table 21: BDD performance analysis across single attributes either visual or demographic based on intensity bias

| Attribute | | FRCNN | | | | YOLO | | | |
| --- | --- | --- | --- | --- | --- | --- | --- | --- | --- |
| Attr Group | Attr Value | AP | Precision | Recall | F1 | AP | Precision | Recall | F1 |
| Skin tone | Unknown | 0.22 | 0.71 | 0.25 | 0.35 | 0.10 | 0.69 | 0.12 | 0.17 |
| Skin tone | DS | **0.45** | 0.91 | 0.45 | 0.60 | **0.45** | 0.96 | 0.45 | 0.61 |
| Skin tone | LS | **0.38** | 0.82 | 0.39 | 0.53 | **0.43** | 0.95 | 0.44 | 0.60 |
| Contrast | high | **0.23** | 0.72 | 0.26 | 0.37 | **0.12** | 0.75 | 0.14 | 0.21 |
| Contrast | low | 0.22 | 0.71 | 0.25 | 0.36 | 0.11 | 0.71 | 0.14 | 0.20 |
| Intensity | dark | **0.23** | 0.72 | 0.26 | 0.37 | 0.11 | 0.72 | 0.14 | 0.20 |
| Intensity | light | 0.22 | 0.70 | 0.24 | 0.33 | 0.11 | 0.73 | 0.14 | 0.20 |
| Size | small | 0.23 | 0.71 | 0.25 | 0.36 | 0.11 | 0.72 | 0.14 | 0.20 |

Table 22: MIAP performance analysis across single attributes either visual or demographic based on intensity bias.

| Attribute | | FRCNN | | | | YOLO | | | |
| --- | --- | --- | --- | --- | --- | --- | --- | --- | --- |
| Attr Group | Attr Value | AP | Precision | Recall | F1 | AP | Precision | Recall | F1 |
| Gender | Predominantly Feminine | 0.33 | 0.80 | 0.35 | 0.49 | **0.56** | 0.88 | 0.59 | 0.71 |
| Gender | Predominantly Masculine | **0.38** | 0.79 | 0.40 | 0.53 | 0.54 | 0.88 | 0.57 | 0.69 |
| Gender | Unknown | **0.12** | 0.31 | 0.23 | 0.14 | **0.12** | 0.40 | 0.27 | 0.21 |
| Age | Middle | 0.36 | 0.79 | 0.38 | 0.51 | 0.54 | 0.88 | 0.57 | 0.69 |
| Age | Older | **0.38** | 0.80 | 0.39 | 0.53 | **0.63** | 0.89 | 0.67 | 0.76 |
| Age | Unknown | **0.08** | 0.26 | 0.21 | 0.11 | **0.07** | 0.31 | 0.21 | 0.13 |
| Age | Young | 0.35 | 0.83 | 0.36 | 0.51 | 0.56 | 0.87 | 0.60 | 0.71 |
| Skin tone | DS | **0.27** | 0.43 | 0.36 | 0.31 | **0.46** | 0.59 | 0.66 | 0.62 |
| Skin tone | LS | **0.27** | 0.48 | 0.35 | 0.33 | **0.46** | 0.63 | 0.65 | 0.64 |
| Skin tone | Unknown | 0.24 | 0.53 | 0.30 | 0.32 | 0.29 | 0.69 | 0.37 | 0.48 |
| Contrast | high | **0.28** | 0.60 | 0.33 | 0.40 | **0.38** | 0.72 | 0.47 | 0.57 |
| Contrast | low | 0.22 | 0.47 | 0.30 | 0.28 | 0.30 | 0.62 | 0.42 | 0.50 |
| Intensity | dark | 0.24 | 0.51 | 0.31 | 0.32 | 0.30 | 0.65 | 0.40 | 0.49 |
| Intensity | light | **0.25** | 0.51 | 0.31 | 0.32 | **0.35** | 0.66 | 0.46 | 0.54 |
| Size | large | 0.18 | 0.29 | 0.30 | 0.17 | **0.38** | 0.51 | 0.64 | 0.57 |
| Size | small | **0.27** | 0.62 | 0.31 | 0.39 | 0.33 | 0.74 | 0.39 | 0.51 |

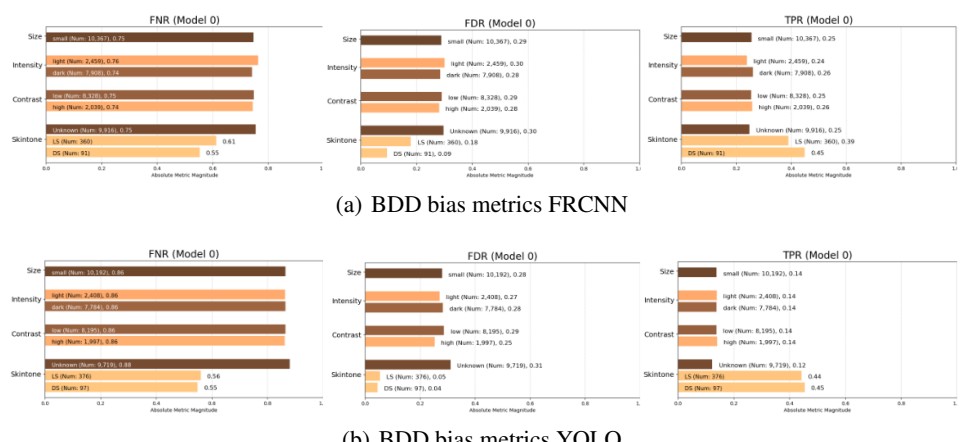

(a) BDD bias metrics FRCNN

(b) BDD bias metrics YOLO

Figure 14: Experiment 4 bias metrics visualization for BDD while injecting intensity biases into the object detector. Photo courtesy of author.

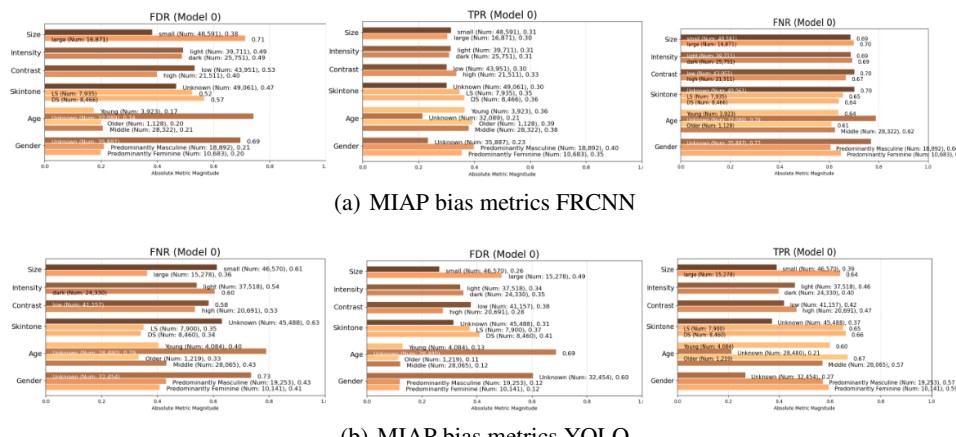

(a) MIAP bias metrics FRCNN

(b) MIAP bias metrics YOLO

Figure 15: Experiment 4 bias metrics visualization for MIAP while injecting intensity biases into the object detector. Photo courtesy of author.

### B.4.4 INTENSITY BIAS FAIRNESS

Table 23: Fairness analysis of BDD dataset in fourth experiment with reference group of unknown skin tone individuals with low contrast, low intensity, and small size. Intensity bias is injected to the detector.

| | Attribute | YOLO | | | | FRCNN | | | |
|---|---|---|---|---|---|---|---|---|---|
| Attr Group | Attr Value | SP | EO | EOP | FTA | SP | EO | EOP | FTA |
| Skin tone | DS | | | | ✓ | | | | ✓ |
| Skin tone | LS | | | | ✓ | | | | ✓ |
| Skin tone | Unknown | ✓ | ✓ | ✓ | | ✓ | ✓ | ✓ | |
| Contrast | high | | ✓ | ✓ | | | ✓ | ✓ | |
| Contrast | low | ✓ | ✓ | ✓ | | ✓ | ✓ | ✓ | |
| Intensity | dark | ✓ | ✓ | ✓ | | ✓ | ✓ | ✓ | |
| Intensity | light | | ✓ | ✓ | | | ✓ | ✓ | |

Table 24: Fairness analysis of MIAP dataset with reference group of Middle-aged male individuals with unknown skin tone, low-contrast, low intensity, and small size. Intensity bias is injected to the detector.

| | Attribute | FRCNN | | | | YOLO | | | |
|---|---|---|---|---|---|---|---|---|---|
| Attr Group | Attr Value | SP | EO | EOP | FTA | SP | EO | EOP | FTA |
| Gender | Predominantly Feminine | | ✓ | ✓ | | | ✓ | ✓ | ✓ |
| Gender | Predominantly Masculine | ✓ | ✓ | ✓ | ✓ | ✓ | ✓ | ✓ | ✓ |
| Gender | Unknown | | | | | | | | |
| Age | Middle | ✓ | ✓ | ✓ | | ✓ | ✓ | ✓ | ✓ |
| Age | Older | | ✓ | ✓ | | ✓ | ✓ | ✓ | ✓ |
| Age | Unknown | | | | | | | | |
| Age | Young | | ✓ | ✓ | | ✓ | ✓ | ✓ | ✓ |
| Skin tone | DS | | ✓ | ✓ | ✓ | | | | ✓ |
| Skin tone | LS | | ✓ | ✓ | | | | | ✓ |
| Skin tone | Unknown | ✓ | ✓ | ✓ | | ✓ | ✓ | ✓ | |
| Contrast | high | | ✓ | ✓ | | | ✓ | ✓ | |
| Contrast | low | ✓ | ✓ | ✓ | | ✓ | ✓ | ✓ | |
| Intensity | dark | ✓ | ✓ | ✓ | ✓ | ✓ | ✓ | ✓ | |
| Intensity | light | | ✓ | ✓ | | | ✓ | ✓ | |

