# OpenReview forum: "Person Detection Through the Lens of Algorithmic Bias"
_ICLR.cc/2025/Conference — Submitted to ICLR 2025_

### Official Review · Reviewer_BR47 · 2024-10-17

**Soundness:** 1
**Presentation:** 3
**Contribution:** 2
**Rating:** 3
**Confidence:** 4

**Summary:**

This paper identifies the prevalence of sampling and label bias within popular benchmark datasets, in particular age, gender, and racial bias in person detection datasets. The authors acknowledge the importance of robustness towards blurriness, contrast and intensity in image data with respect to overall detection performance and also bias metrics. Using a variety of detection performance metrics, and fairness metrics, the authors attempt to answer the two following questions:
1. do common real-world image manipulation techniques (blurring and grayscale conversion) impact fairness?
2. does a model biased towards visual features such as size, contrast, and brightness impacts fairness on different demographic groups?

Under four experimental settings, the authors conduct a number of experiments and produce multiple quantitative results to answer these two questions, and conclude that blurring, grayscale conversion, and a biased model do indeed exhibit worse fairness. Based on these findings, the authors call for closer attention towards producing unbiased models and alleviating bias within real-world scenarios.

**Strengths:**

In general, this paper seeks to answer very important questions and the motivation behind the paper is well supported in other literature, which is well framed by the literature review.

The tables and figures for ap / fdr, fnr, tpr are good. The qualitative results in the figures in the appendix clearly show that on the whole, YOLO has less bias compared to Faster-RCNN, and that there is a disparity in performance given certain characteristics. The tables in the appendix analysing fairness clearly illustrate the disparity in fairness between YOLO and Faster-RCNN, and across different attributes.

Fig 1-3 cleary illustrate the methodology in section 5 of the body of the paper. Apart from a few formatting issues and typos, the figures are clear, sufficiently captioned, and are well supported by the text.

The results section gives a comprehensive description of both the quantitative and qualitative results presented in the figures and tables in the appendix.

**Weaknesses:**

## Main Concerns
---
My primary concern is the analysis and interpretation of the results. Minimal attempt is made to compare the performance between the four different experimental settings. The authors state in section 6.2 (top of page 7):

"Notably, individuals with dark and light skin tones do not meet any fairness measures compared to the reference group of unknown skin tone. All other attribute groups retain the same performance as Experiment 1."
"Interestingly, in contrast to Experiment 1, both detectors failed FTA for all groups and attributes. See Table 8 and Table 4 for a visual comparison."

These are interesting findings and need to be expanded upon. I believe these are the only two comments that draw comparisons between the different experimental settings. To answer the two research questions stated in the paper, I believe there should be more comparisons with the baseline experiment how the image data manipulation (experiment setting 2) and injected model bias (experiment setting 3) affect fairness. Without comparing these experimental settings, we cannot know how the settings affect fairness, unless I have misunderstood, in which case please explain why comparing the experimental settings is not important.

Following from this, it is expected and known that test/inference on a different distribution to training leads to worse performance. It does not necessarily follow that the reduction in overall performance / fairness metrics after image manipulation (experiment 2 compared to experiment 1) is an indicator of bias. For instance, how much does performance in one demographic drop compared to another? Has performance between the different demographics widened or narrowed? In fact, the author's results suggest that for Faster-RCNN, the performance gaps between characteristics narrow (e.g. for MIAP under experiment 1, Faster-RCNN achieves 0.63 on DS, and 0.59 on LS, a range of 0.04 on non-unknown demographics. Under experiment 2, Faster-RCNN achieves 0.27 on both DS and LS, a range of 0). Image manipulation affects one demographic more than another, but in the image-manipulated domain the performance is more equal. It is therefore not obvious how to interpret the results. Some comparison between the experimental settings is necessary however. Additionally it might be beneficial to visualise this impact qualitatively, and visualise object detection performance before and after image manipulation. Methodology of Majumdar et al. [1] could be useful here, using ablation-CAM to determine how masking regions related to certain racial or gender phenotypes (e.g. hairstyles) affects object detection.

Section 7.2 discusses the results with respect to answering research question 1 (how does biased data impact fairness?). However, the discussion largely describes how AP and recall drop when testing on biased data. This does not satisfactorily answer question 1, and again, insufficient comparison is made between experiment 1 and experiment 2 to anwer this research question. In general, section 7 does not discuss fairness and therefore do not sufficiently answer the proposed research questions.

On page 6, section 6.1 the authors state that 'female individuals do not satisfy SP for either detector' while 'male individuals pass all fairness tests for both detectors.' It is not clear how males can pass, and females can fail, given the provided definition for SP: 'A classifier is deemed fair if the likelihood of a favorable outcome is the same for both privileged and unprivileged groups.' Similarly, it is not clear how the 'unknown' demographic is treated with respect to these fairness metrics. Again, how can the male group pass SP while the unknown group fails.

---
## The methodology omits details in certain areas:
---
I could not find a description of how bias is injected into the nms step for experiment 3 indicated in section 5.5.

There is no description of the metrics used for fairness through awareness. Verma et al. [2] caution readers that the distance metric must be chosen with care. It is hard to conlude therefore whether the findings from this paper are reasonable without knowledge of the distance metrics used. The only information is that aequitas toolkit is used, which offers no insight into fairness through awareness.

It is not clear if the models are re-trained on the biased dataset. I assume not but this is not indicated in the paper.

In section 6.1 the authors comment that 99% of the people in the BDD dataset have unknown skin tone. There is no description of the number of instances of each category. Are there enough instances of LS and DS demographics to draw meaningful conclusions?

---
## Other issues:
---
The authors correctly identify the prevalence of blurring in real-world scenarios. It would be nice to have at least a reference to compression (e.g. commonly used JPEG), which has significant overlap with blurring and also degrades colour information in an image which has been shown to affect racial bias in deep learning models. Dodge and Karam (2016) [3] provide some analysis and might be a useful reference.

There is no reference to the recent FACET [4] paper, which analyses fairness on person image dataets. Although the FACET paper does not consider the low-quality image setting, it should be referred to.

In section 7.1, the authors state 'YOLO may also have benefited from higher image quality within MIAP.' What do the authors mean by this? Is the comparison between Faster-RCNN and YOLO performance fair?

---
## I noticed a few issues that can be trivially corrected and did not influence my rating:
---
- p2&3, section 3.3: formatting issues around dataset referencing.
- description of MIAP as 'more inclusive images for people.' This occurs in multiple places, and should be changed to 'more inclusive annotations for people.'
- references throughout are doubled up, such that authors names appear twice (e.g. 3.3. page 3: 'wilson et al. wilson et al.' and 3.3 page 3: 'Balakrishnan et al. Balakrishnan et al.' - page 3). Punctuation around names referenced in the body of the text is inconsistent throughout.
- p8, 6.3.2. (¿0.8 AP).
- 7.1: 'potentialyl'
- 3.3: 'lends' instead of 'lens' and in this same sentence, remove the word 'on'
- fig. 2 caption - adding blurriness and grayscale conversion TO a picture?
- fig. 3 boxes are not equal widths / spacing. looks messy.

---
## References
---
- [1] Puspita Majumdar, Surbhi Mittal, Richa Singh, Mayank Vatsa: Unravelling the Effect of Image Distortions for Biased Prediction of Pre-trained Face Recognition Models. ICCVW 2021: 3779-3788
- [2] Sahil Verma, Julia Rubin: Fairness definitions explained. FairWare@ICSE 2018: 1-7
- [3] Samuel Fuller Dodge, Lina J. Karam: Understanding how image quality affects deep neural networks. QoMEX 2016: 1-6
- [4] Laura Gustafson, Chloé Rolland, Nikhila Ravi, Quentin Duval, Aaron Adcock, Cheng-Yang Fu, Melissa Hall, Candace Ross: FACET: Fairness in Computer Vision Evaluation Benchmark. ICCV 2023: 20313-20325

**Questions:**

Without comparing experimental settings, we cannot know how the settings affect fairness, unless I have misunderstood, in which case please explain why comparing the experimental settings is not important?

How can males pass SP, and females fail SP, given the provided definition for SP: 'A classifier is deemed fair if the likelihood of a favorable outcome is the same for both privileged and unprivileged groups.'

The authors comment that 99% of the people in the BDD dataset have unknown skin tone. There is no description of the number of instances of each category. Are there enough instances of LS and DS demographics to draw meaningful conclusions?

In section 7.1, the authors state 'YOLO may also have benefited from higher image quality within MIAP.' What do the authors mean by this? Is the comparison between Faster-RCNN and YOLO performance fair?

---

### Official Review · Reviewer_f3dz · 2024-10-23

**Soundness:** 3
**Presentation:** 3
**Contribution:** 3
**Rating:** 3
**Confidence:** 5

**Summary:**

The paper titled "Person Detection Through the Lens of Algorithmic Bias" is a comprehensive study that investigates the biases present in machine learning models and datasets used for person detection, particularly in safety-critical applications like autonomous vehicles and security monitoring. The authors focus on the popular benchmark datasets MIAP (More Inclusive Images for People) and BDD (Berkeley DeepDrive), which are known to suffer from sampling and labeling biases. The study conducts an in-depth analysis of both dataset and model biases, with a novel examination of person detection in low-quality or crowded images. The key findings indicate that real-world image manipulations, such as blurriness, and model reliance on features like contrast or brightness significantly impact fairness across race, gender, and age demographics. The paper contributes to the field by providing insights into how biases can be mitigated to design more robust object detection models.

**Strengths:**

1. The paper offers a novel analysis of bias in real-world image datasets, focusing on the impact of image quality and model features on detection fairness.

2. The study employs a rigorous methodology, using two datasets (MIAP and BDD) and two object detectors (Faster RCNN and YOLO), and introduces controlled biases to evaluate their impact.

3. The paper utilizes established fairness metrics to evaluate the performance of object detection techniques, providing a quantitative measure of bias.

**Weaknesses:**

1. While the study provides valuable insights, it is limited to two datasets and two object detectors.  The authors should discuss the generalizability of their findings to other datasets and models.

2. The paper could benefit from an analysis of how dataset size and diversity affect the biases observed.

3. While the paper identifies biases, it could further strengthen its contribution by proposing and evaluating specific mitigation strategies or algorithms to reduce these biases.

4. A more in-depth discussion on the ethical implications of algorithmic bias in safety-critical applications would enhance the paper's contribution to the field.

5. The paper could benefit from a comparison with the latest state-of-the-art methods in bias detection and mitigation to position its contributions within the current research landscape.

**Questions:**

Please refer to Section Weakness.

---

### Official Review · Reviewer_T66J · 2024-10-29

**Soundness:** 2
**Presentation:** 3
**Contribution:** 2
**Rating:** 3
**Confidence:** 3

**Summary:**

This work studies the relation between dataset bias, model bias, and performance fairness in person detection, which is an important topic for real-world application scenarios, such as autopilot and security monitoring. This work conducts thorough experiments and analysis to the image manipulations and finds the negative effects on fairness for various subgroups.

**Strengths:**

a)	This fairness and bias topic is import for person detection. It could be beneficial to future model design and dataset construction in this research field.

b)	Extensive experiments are conducted on various datasets and person detection methods to analyze the algorithmic bias.

**Weaknesses:**

i.	The work only evaluates the fairness on two person detection datasets, but there are many popular pedestrian datasets, such as Caltech and CityPersons. The authors should include more datasets to strengthen the persuasiveness of the conclusion.

ii.	This work only conducts experiments on the general detection methods, including YOLO and Faster RCNN, the specially designed pedestrian detectors, such as CSP and PRNet, should also be included.

iii.	Only artificial data biases, such as blurry and grayscale, are used. Real bias datasets, such as NightOwls dataset, should be used to test the fairness in real world scenario.

iv.	The main part of this paper should be self-contained, but this work put all results Figures and Tables in the Appendix.

v.	On page 2, “While most of the works reviewed in Section 2” should be “While most of the works reviewed in Section 3”.

vi.	Figures 4 to 15 are blurry.

**Questions:**

See Weakness

---

### Official Review · Reviewer_huMs · 2024-11-03

**Soundness:** 1
**Presentation:** 1
**Contribution:** 1
**Rating:** 1
**Confidence:** 5

**Summary:**

This paper discussed person detection in terms of model and dataset biases. Two representative object detectors including Mask RCNN and YOLOv3 are evaluated on two popular datasets including MIAP and BDD. Based on the proposed pipeline, several findings are summarized to guide future object detection models.

**Strengths:**

I appreciate that the authors conducted extensive experiments to discuss bias and fairness in object detection.

**Weaknesses:**

Although the authors claim that this is the first time to analyze person detection in low-quality or crowded pictures, the novelty of this paper is limited. First, there are a lot of person detection datasets focusing on these challenges, such as CrowdHuman, TinyPerson or other aerial person detection datasets. Second, the whole manuscript is more like an experiment report rather than academic paper. I am looking forward to boarder theoretical analysis and technical contribution. Third, the selected object detectors (i.e., Mask R-CNN and YOLOv3) are outdated, which is difficult to support the statements. The authors should use DINO based detectors and YOLOv8 above. Fourth, the two evaluated datasets are also not comprehensive. I suggest that the authors add more datasets as mentioned above in the experiment.

On the other hand, it is weird to put every quantitative table in the appendix. Some important results including attribute-wise accuracy on two datasets must be shown in the main text. I suggest that the authors can summarize it in a unified figure for clarity. To this end, the content of the first three sections can be cut down.

The authors highlight three additional visual features including size, contrast and intensity. Can you explain why these factors are special for object detection? How do these factors impact on the model performance more than others in real-world challenges? It would be more convinced if the authors develop an improved object detector based on the findings. For example, how do the new network reduce the influence of above biases to improve accuracy? Thus the technical contribution part can meet the requirement of ICLR.

**Questions:**

Please see the above weaknesses.

---

### Meta-Review · Area_Chair_qb8Z · 2024-12-17

**Metareview:**

This paper presents a study on person detection in the presence of dataset and model bias.

Reviewers find the topic (fairness and bias in object person detection) relevant and important, they find this study not particularly convincing. Criticism includes a small scope of dataset coverage (only two datasets are included, and other relevant datasets (e.g., Caltech and CityPersons, Reviewer T66J) are skipped. The detectors' selection is also limited, and only artificially added biases are studied (e.g., blur). There is an opportunity to study real-world biases (e.g., Rev. T66J points out the NightOwls dataset).

While reviewers appreciate the efforts and find the topic relevant, they conclude that the study is not sufficiently well-executed to pass the bar. As overall ratings were 1, 3, 3, 3, and there was no rebuttal, I do not recommend accepting this paper.

**Additional Comments On Reviewer Discussion:**

There was no author response or discussion.

---

### Decision · Program_Chairs · 2025-01-22

Reject